

# The last interglacial (MIS 5e) cycle at Little Bahama Bank:
# A history of climate and sea-level changes

Anastasia Zhuravleva[1] and Henning A. Bauch[2]

[1]Academy of Sciences, Humanities and Literature, Mainz, c/o GEOMAR Helmholtz Centre for Ocean Research, Wischhofstrasse 1-3, Kiel, 24148, Germany
[2]Alfred Wegener Institute, Helmholtz Centre for Polar and Marine Research c/o GEOMAR Helmholtz Centre for Ocean Research, Wischhofstrasse 1-3, Kiel, 24148, Germany

*Correspondence to*: Anastasia Zhuravleva (azhuravleva@geomar.de)

**Abstract.** Shallow-water sediments of the Bahama region containing the last interglacial (MIS 5e) are ideal to investigate the region's sensitivity to past climatic and sea level changes. Here we present new faunal, isotopic and XRF-sediment core data from the northern slope of the Little Bahama Bank. The results suggest that the bank top remained flooded across the last interglacial "plateau", ~129-117 ka, arguing for a relative sea level above -6 m for this time period. In addition, climatic variability, which today is closely coupled with movements of the intertropical convergence zone (ITCZ), is interpreted based on stable isotopes and foraminiferal assemblage records. During early MIS 5e, the mean annual ITCZ position moved northward in line with increased solar forcing and a recovered Atlantic Meridional Overturning Circulation (AMOC). The early MIS 5e warmth peak was intersected, however, by a millennial-scale cooling event, consistent with a southward shift in the mean annual ITCZ position. This tropical shift is ascribed to the transitional climatic regime of early MIS 5e, characterized by persistent high-latitude freshening and, thereby, unstable AMOC mode. Our records from the Bahama region demonstrate that not only was there a tight relation between local sedimentation regimes and last interglacial sea level history, via the atmospheric forcing we could further infer an intra-interglacial connectivity between the polar and subtropical latitudes that left its imprint also on the ocean circulation.

## 1 Introduction

The last interglacial (MIS 5e) has attracted a lot of attention as a possible analog for future climatic development as well as a critical target for validation of climatic models. This globally warmer-than-preindustrial interval is associated with significantly reduced ice sheets and a sea level rise up to 6-9 meters above the present levels (Dutton et al., 2015; Hoffman et

al., 2017). However, controversy still exists regarding the initiation and duration of the sea level highstand as well as about

any sea level variability within that time period (Hearty et al., 2007; Kopp et al., 2009; Grant et al., 2012; Masson-Delmotte

et al., 2013). Also, the spatial coverage of the existing sea surface temperature (SST) reconstructions is insufficient to allow

for a robust understanding of the climatic forcings at play during the last interglacial.

At the western boundary of the wind-driven subtropical gyre (STG), sediments from the shallow-water carbonate platforms of

the Bahama archipelago can serve as an important climatic link between the tropical and subpolar N. Atlantic. Given its critical

location near the origin of the Gulf Stream, sediment records from the Bahama Bank region have been investigated in terms

of climatic variability, ocean circulation, sea level change and sediment diagenesis (Slowey and Curry, 1995; Henderson et

al., 2000; Slowey et al., 2002; Roth and Reijmer, 2004; 2005; Chabaud, 2016). For the last interglacial, earlier studies were

focused on absolute dating of corals and sediments (Neumann and Moore, 1975; Slowey et al., 1996; Henderson and Slowey,

2000) as well as on coastal geomorphology and changes in sediment properties in relation to sea level fluctuations (Hearty and

Neumann, 2001; Lantzsch et al., 2007; Chabaud et al., 2016). However, a thorough study of the last interglacial climate

evolution underpinned by a critical stratigraphical insight deduced from periplatform oozes is lacking so far, as previous

authors mainly worked on timescales covering several interglacials (Lantzsch et al., 2007; Chabaud et al., 2016). Here, we

attempt to close this gap, by using core MD99-2202 from the upper northern slope of the Little Bahama Bank (LBB), which

is the northernmost shallow-water carbonate platform of the Bahamian archipelago.

**2 Regional Setting**

Core MD99-2202 (27°34.5´ N, 78°57.9´ W, 460 m water depth) was taken from the northern slope of the LBB in the vicinity

to the western boundary current of the N. Atlantic STG, the Gulf Stream (Fig. 1A). The Gulf Stream supplies both heat and

salt to the high northern latitudes and thereby constituting the upper cell of the Atlantic Meridional Overturning Circulation

(AMOC). It originates from the Florida Current after it leaves the Gulf of Mexico through the Straits of Florida and merges

with the Antilles Current to the north of the LBB.

In the western subtropical N. Atlantic two distinctly different layers can be distinguished within the upper 500 m of the water

column. The uppermost mixed layer (upper 50-100 m) is occupied by warm and comparatively fresh waters (T>24°C, S<36.4

psu), predominantly coming from the equatorial Atlantic (Schmitz and McCartney, 1993; Johns et al., 2002). Properties of this



water mass vary significantly on seasonal timescales and are closely related to the latitudinal migration of the intertropical

convergence zone (ITCZ) (Fig. 1B-C). During boreal winter (December-April), when the ITCZ is in its southernmost position,

the Bahama region is dominated by relatively cool, stormy weather with prevailing northern and northeastern trade winds and

is affected by cold western fronts, that increase evaporation and vertical convective mixing (e.g., Wilson and Roberts, 1995).

During May to November, as the ITCZ moves northward, the LBB is influenced by relatively weakened trade winds from the

east and southeast, increased precipitation and a very warm pool of waters (T >28°C) which expands into the Bahama region

from the Caribbean Sea and the equatorial Atlantic (Stramma and Schott, 1999; Wang and Lee, 2007; Levitus et al., 2013).

Today the LBB region lies at the northern edge of the influence of the tropical pool waters, making our site particularly

sensitive to monitor past shifts of the ITCZ. The mixed layer is underlain by a thermocline layer, which is comprised of a

homogeneous pool of comparatively cool and salty (12<T<24°C, S>36.4 psu) water (Schmitz and Richardson, 1991). These

"mode" waters are formed in the N. Atlantic STG through wintertime subduction of surface waters driven by wind-driven

Ekman downwelling and buoyancy flux (Slowey and Curry, 1995).

Along the slopes of the LBB, sediments are composed from varying amounts of sedimentary inputs from the platform top and

from the open ocean, depending on the global sea level state (Schlager et al., 1994). During interglacial highstands, when the

platform top is submerged, the major source of sediment input is the downslope transport of fine-grained aragonite needles,

precipitated on the platform top. This material incorporates significantly higher abundances of strontium (Sr), than found in

pelagic-derived aragonite (e.g., pteropods) and calcite material from planktic foraminifera and coccoliths (Morse and

MacKenzie, 1990). As in a periplatform interglacial environment modifications of aragonite content due to sea floor

dissolution as well as by winnowing of fine-grained material are minimal (Schlager et al., 1994; Slowey et al., 2002; Chabaud

et al., 2016), thick sediment packages are accumulated on the slopes of the platform, producing high resolution records of

interglacial climate (Roth and Reijmer, 2004; 2005). During glacial lowstands on the contrary, as the bank top is exposed,

aragonite production is limited, sedimentation rates are strongly reduced and coarser-grained consolidated sediments are

formed from the pelagic organisms (Slowey et al., 2002; Lantzsch et al., 2007).

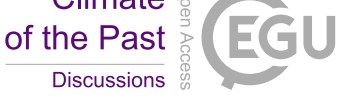

## 3 Methods

### 3.1 Foraminiferal counts and stable isotopes analyses

Planktic foraminiferal assemblages were counted on representative splits of the 150-250 µm fraction containing at least 300 individual specimens. Counts were also performed in the >250 µm fraction. The census data from the two size fractions were added up and recalculated into relative abundance of planktic foraminifera in the fraction >150 µm. Faunal data were obtained at each 2 cm for the core section between 508.5 and 244.5 cm and at each 10 cm between 240.5 and 150.5 cm. According to a

standard practice, *Globorotalia menardii* and *Globorotalia tumida* as well as *Globigerinoides sacculifer* and *Globigerinoides trilobus* were grouped together (Poore et al., 2003; Kandiano et al., 2012; Chabaud, 2016), and referred to as *G. menardii* and *G. sacculifer*, respectively.

New oxygen isotope data were produced at 2 cm steps using ~10-30 tests of *Globorotalia truncatulinoides* (dex) and ~5-20 tests of *Globorotalia inflata* for depths 508.5-244.5 cm and 508.5-420.5 cm, respectively. Analyses were performed using a

Finnigan MAT 253 mass spectrometer at the GEOMAR Stable Isotope Laboratory. Calibration to the Vienna Pee Dee Belemnite (V-PDB) isotope scale was made via the NBS-19 and an internal laboratory standard. The analytical precision of in-house standards was better than 0.07‰ (1σ) for $\delta^{18}O$. Deep-dwelling foraminifera *G. truncatulinoides* and *G. inflata* are found in greatest abundances at the base of the seasonal thermocline (100-200 m), under environmental stress, e.g., temperatures warmer than 16°C, however, the species can migrate to greater depths (Cléroux et al., 2007). As calcification of

their tests starting in the mixed layer continues in the main thermocline, the abovementioned species are thought to accumulate hydrographic signals from different water depths (Groeneveld and Chiessi, 2011; Mulitza et al., 1997).

### 3.2 XRF scanning

X-ray fluorescence (XRF) analysis was performed in two different runs using the Aavatech XRF Core Scanner at Christian-Albrecht University of Kiel (for technical details see Richter et al., 2006). To obtain intensities of elements with lower atomic

weight (e.g., calcium (Ca), chlorine (Cl)), XRF scanning measurements were carried out with the X-ray tube voltage of 10 kv, the tube current of 750 µA and the counting time of 10 seconds. To analyze heavy elements (e.g., iron (Fe), Sr), the X-ray generator setting of 30 kv and 2000 µA and the counting time of 20 seconds were used; a palladium thick filter was placed in the X-ray tube to reduce the high background radiation generated by the higher source energies. XRF Core Scanner data were

collected directly from the split core sediment surface, that had been flattened and covered with a 4 µm-thick ULTRALENE

SPEXCerti Prep film to prevent contamination of the measurement unit and desiccation of the sediment (Richter et al., 2006;

Tjallingii et al., 2007). The core section between 150 and 465 cm was scanned at 3 mm step size, whereas the coarser-grained

interval between 465 and 600 cm was analyzed at 10 mm resolution.

To account for potential biases related to physical properties of the sediment core, XRF intensities of Sr were normalized to

Ca (Fig. 2), the raw total counts of Fe and Sr were normalized to the total counts of the 30kv-run; counts of Ca and Cl were

normalized to the total counts of 10kv-run, excluding Rh intensity, because this element intensities are biased by the signal

generation (Bahr et al., 2014).

All data will be made available in the online database PANGAEA (www.pangaea.de).

## 4 Age model

By using our foraminiferal assemblage data, we were able to refine the previously published age model of core MD99-2202

(Lantzsch et al., 2007). To correctly frame MIS 5e, stratigraphic subdivision of the unconsolidated aragonite-rich sediment

package between 190 and 464 is essential (Fig. 3). In agreement with Lantzsch et al. (2007) we interpret this core section to

comprise MIS 5, which is supported by key biostratigraphic markers used to identify the well-established faunal zones of late

Quaternary (Ericson and Wollin, 1968). Thus, the last occurrences of *G. menardii* and *G. menardii* flexuosa at the end of the

aragonite-rich sediment package are in agreement with the estimated late MIS 5 age (ca. 80-90 ka; Boli and Saunders, 1985;

Slowey et al., 2002; Bahr et al., 2011; Chabaud, 2016). The coherent variability observed between aragonite content and

relative abundances of warm surface-dwelling foraminifera of *Globigerinoides* genus (*G. ruber,* white and pink varieties, *G.*

*conglobatus* and *G. sacculifer*) between ~200-300 cm, points to simultaneous climate and sea level-related changes and likely

reflects the warm/cold substages of MIS 5. The detected substages were then correlated with the global isotope benthic stack

LS16 (Lisiecki and Stern, 2016) using AnalySeries 2.0.8 (Paillard et al., 1996). Further, boundaries between MIS 6/5e and

5e/5d as well as the penultimate glaciation (MIS 6) peak, defined from $\delta^{18}O$ record of *G. ruber* (white), were aligned to the

global benthic stack (Lisiecki and Stern, 2016).

Given that sedimentation rates at the glacial/interglacial transition could have changed drastically due to increased production

of Sr-rich aragonite material above the initially flooded carbonate platform top (Roth and Reijmer, 2004; Chabaud et al., 2016),

we applied an additional age marker to better frame the onset of the MIS 5e "plateau" and to allow for a better core-to-core

comparison. Thus, we tied the increased relative abundances of warm surface-dwelling foraminifera of *Globigerinoides* genus,

which coincides with the rapid decrease in foraminiferal $\delta^{18}$O record at 456 cm, with the onset of MIS 5e "plateau" at ~129 ka

(Masson-Delmotte et al., 2013). This age is in good agreement with many marine and speleothem records, dating a rapid post-

stadial warming and monsoon intensification to 129-128.7 ka (Galaasen et al., 2014; Govin et al., 2015; Jiménez-Amat and

Zahn, 2015) coincident with the sharp methane increase in the EPICA Dome C ice core (Loulergue et al., 2008; Govin et al.,

2012). Although we do not apply a specific age marker to frame the decline of the MIS 5e "plateau", the resulting decrease in

the percentage of warm surface-dwelling foraminifera of *Globigerinoides* genus as well as the initial increase in the planktic

$\delta^{18}$O values dates back to ~117 ka (Fig. 4), which broadly coincides with the cooling over Greenland (NGRIP community

members, 2004). A similar subtropical-polar climatic coupling was proposed in earlier studies from the western N. Atlantic

STG (e.g., Vautravers et al., 2004; Schmidt et al., 2006a; Bahr et al., 2013; Deaney et al., 2017).

## 5    Results

### 5.1 XRF data in the lithological context

In Fig. 2 XRF-derived elemental data are plotted against lithological and physical sediment properties. Beyond the intervals

with low Ca counts corresponding to high Cl intensities (at 300-325 cm and 395-440 cm), Ca intensities do not vary

significantly, which is in line with a stable carbonate content of about 94% Wt (Lantzsch et al., 2007). Our Sr record closely

follows the aragonite curve and the grain size data, demonstrating that the interglacial minerology is dominated by aragonite.

Beyond the intervals containing reduced Ca intensities, a good coherence between Sr/Ca and aragonite content is observed.

The rapid increase in Sr/Ca and aragonite is found at the end of the penultimate deglaciation (Termination 2, T2), coeval with

the elevated absolute abundances of *G. menardii* per sample (Fig. 4). The gradual step-like Sr/Ca and aragonite decrease

characterizes both the glacial inception and the later MIS 5 phase. Intensities of Fe abruptly decrease at the beginning of the

last interglacial, but gradually increase during the glacial inception (Fig. 5D). At ~120 ka (355 cm), a minor but clear increase

in Sr intensities goes along with the change in aragonite and grain-size (Figs. 2 and 4), arguing that this feature is not a signal

artefact but represents a significant sedimentological shift. Between ~112 and 114.5 ka, the actual XRF measurements were

affected by a low sediment level in the core tube.





## 5.2 Climate-related proxies

During the major deglacial transition, ~135-129 ka, low isotopic gradients between *G. ruber* (white), *G. truncatulinoides* (dex)

and *G. inflata* are consistent with high relative abundances of *G. truncatulinoides* (dex) and *G. inflata* (Fig. 5). Across MIS 5e

species of *Globigerinoides* genus dominate the total assemblage, however, significant changes in the proportions of three main

*Globigerinoides* species are observed: *G. sacculifer* and *G. ruber* (pink) essentially dominate the assemblage during early MIS

5e (129-124 ka), whereas *G. ruber* (white) proportions are at their maximum during late MIS 5e (124-117 ka). At the onset of

MIS 5e, *G. ruber* (pink) and *G. sacculifer* relative abundances rise in a successive manner, with a rapid increase in *G. sacculifer*

occurring c. 2 ka later than the rise in *G. ruber* (pink) proportions. At around 127 ka, all $\delta^{18}O$ records abruptly increase, *G.*

*falconensis* and *G. inflata* reappear, while relative abundances of *G. ruber* (pink) and *G. sacculifer* become reduced (Figs. 5-

6). After 120 ka, $\delta^{18}O$ values in *G. ruber* (white) and *G. truncatulinoides* (dex) become unstable (Fig. 5A-B). This instability

coincides with an abrupt drop in *G. sacculifer* relative abundances (Fig. 6B).

## 6    Discussion

### 6.1 Platform sedimentology and sea level change

In shallow-water records from the Bahamas, downcore variations in Sr/Ca intensity ratio can be applied as a good proxy for

relative sea level (RSL) change (Chabaud et al., 2016). However, given that the measured intensities of Ca account for more

than 70% of all elements signal intensities, Sr/Ca values are strongly dependent on the quality of the Ca signal. While our Sr

record likely represents a non-affected signal because of good coherence with the aragonite curve, some of the Ca intensity

values are reduced due increased seawater content, as evidenced by simultaneously measured elevated Cl intensities (Fig. 2).

Because enhanced seawater content in the sediment appears to reduce only Ca intensities, leaving measures of elements with

higher atomic numbers (e.g., Fe, Sr) less affected (Tjallingii et al., 2007; Hennekam and de Lange, 2012), normalization of Sr

counts to Ca results in very high Sr/Ca intensity ratios across the Cl-rich intervals. The general consistency of the measured

Sr intensities argues against an early marine diagenesis that would strongly reduce and homogenize the Sr/Ca intensity ratio,

altering isotopic signatures and often causing a change in sediment color (Chabaud, 2016) that is not observed in our sediment

core. Consequently, the spikes in Sr/Ca within the last interglacial could be related to a change in physical properties of the

sediment, such as elevated sediment porosity linked to increased sand size fraction, that would facilitate enhanced pore-water

content in these intervals (Henderson et al., 2000; Tucker and Bathurst, 2009; Chabaud, 2016). Indeed, the late MIS 5e Sr/Ca

spike falls within the interval of strongly decreased sediment density and increased porosity (Labeyrie and Reijmer, 2005).

However, on the basis of these data alone it is not possible to assess whether the observed sedimentological and geochemical

shifts represent a syn- or postdepositional change. Yet, comparing all the records it seems conceivable that the pronounced

Sr/Ca and Cl spikes may contain clues about interglacial sedimentary regime changes on the upper slope of the LBB at these

times.

Beyond these problematic intervals described above, XRF-derived Sr/Ca values agree well with the aragonite curve and, thus,

can be interpreted in terms of RSL variability (Fig. 4). Around 129 ka, Sr/Ca as well as aragonite content rapidly reach

maximum values, indicating the onset of the LBB flooding interval. Absolute abundance of *G. menardii* per sample supports

the inferred onset of the flooding interval (Fig. 4D), since amounts of planktic foraminifera in the sample can be used to assess

the relative accumulation of platform-derived vs. pelagic sediment particles (Slowey et al., 2002). After *G. menardii*

repopulated the tropical waters at the end of the penultimate glaciation (Bahr et al., 2011; Chabaud, 2016), its increased

absolute abundances are found between ~131-129 ka. That feature could be attributed to a reduced input of fine-grained

aragonite at times of partly flooded platform. Consequently, as the platform top became completely submerged, established

aragonite shedding gained over pelagic input, thereby reducing the number of *G. menardii* per given sample.

Despite some possible isostatic subsidence (1-2 m per hundred thousand years), the LBB is generally regarded as tectonically

stable (Carew and Mylroie, 1995; Hearty and Neumann, 2001). Considering that the modern water depth of the platform is

between 6-10 m, a RSL above -6 m of its present position is required to completely flood the platform top and allow for a

drastic increase in aragonite production (Carew and Mylroie, 1997; Chabaud et al., 2016). In that context, the onset of the

major flooding interval with RSL above -6 m could be assumed from c.129 ka on (Fig. 4). Since the last interglacial sea level

highstand is estimated to have been 6-9 meters above the modern (Dutton et al., 2015), an additional sea level rise of 12-16 m

must have been reached some time later within MIS 5e. A late peak is indeed in agreement with a continuing deglaciation

observed in the northern hemisphere (e.g., Bauch et al., 2012; Deaney et al., 2017); a sea–level contribution from the Antarctic

Ice Sheets have also been suggested (Hearty and Neumann, 2001; O´Leary et al., 2013).

Our data do not allow to make assumptions about the exact timing of the last interglacial sea level peak, which is controversially

placed by different studies into either early (Grant et al., 2012; Lisiecki and Stern, 2016), mid or late MIS 5e (Hearty and

Neumann, 2001; Hearty et al., 2007; Kopp et al., 2009; O´Leary et al., 2013; Spratt and Lisiecki, 2016). And yet, our proxy

records suggest that the aragonite production on top of the platform was abundant until late MIS 5e (unequivocally delimited

by foraminiferal $\delta^{18}$O and faunal data), arguing for a longer-lasting flooding interval of the LBB across MIS 5e (~12 ka with

the RSL above -6 m), when compared to previous sea level reconstructions (Fig. 4H). The drop in RSL below -6 m only during

the terminal phase of MIS 5e (~117 ka on our timescale) is corroborated by a coincident changeover in the aragonite content

and the increase in absolute abundance of *G. menardii*, further supporting the hypothesis that aragonite shedding was

suppressed at that time, causing relative enrichment in foraminiferal abundances (per sample).

As mentioned above, the inferred LBB flooding time period differs from the actual minimal ice volume interval, but it appears

to correspond roughly with the well-known last interglacial "plateau" of low benthic $\delta^{18}$O values between ~129 and 116-118

ka (Adkins et al., 1997; Cortijo et al., 1999; Masson-Delmotte et al., 2013). Given that the intra-interglacial sea level change

is plausible, this observation underscores the importance of critical consideration of deep-sea $\delta^{18}$O records, accumulating both

ice volume and ocean temperature/salinity signals.

**6.2 Termination 2**

Prior to the MIS 5e "plateau", the elevated occurrences of transitional to subpolar species *G. inflata* indicate generally cold-

water conditions off the LBB (Fig. 5). Isotopic gradients between $\delta^{18}$O values in surface- and thermocline-dwelling

foraminifera during T2 are strongly reduced, arguing for decreased water column stratification. At times of suppressed

overturning during T2 (Deaney et al., 2017), the inferred decreased stratification could have resulted from sea surface

cooling/salinification and/or subsurface warming (e.g., Zhang, 2007). Nevertheless, direct surface/subsurface temperature

estimations across T2 and early MIS 5e so far reveal warm/cold conditions for the subtropical western N. Atlantic (Bahr et al.,

2013). It is known, however, that species-specific temperature signals should be considered with caution, as they could be

complicated due to adaptation strategies of foraminifera, such as seasonal shifts in the peak foraminiferal tests flux and/or

habitat changes (Schmidt et al., 2006a, b; Bahr et al., 2013; Jonkers and Kučera, 2015). Alternatively, reduced water column

stratification could have led to a situation when calcification of the thermocline-dwelling foraminifera could have commenced

in shallower and, therefore, relatively warmer waters, causing a lower isotopic gradient between shallow- and deep-dwelling foraminifera (Mulitza et al., 1997).

High abundances of *G. truncatulinoides* (Fig. 5E) further support the hypothesis involving reduced stratification and deep vertical mixing, given that today this species requires reduced upper water column stratification to be able to complete its reproduction cycle with changing habitats, from c. 400-600 m to near-surface depths (Lohmann and Schweizer, 1990; Hilbrecht, 1996; Mulitza et al., 1997). For instance, in the modern tropical Caribbean, reproduction of *G. truncatulinoides* is inhibited by strong thermocline in well-stratified waters (Schmuker and Schiebel, 2000). This is in contrast to the subtropical

N. Atlantic where winter sea surface cooling (T<23°C) and deep mixing occur alongside with increase of *G. truncatulinoides* up to 15% (Levitus et al., 2013; Siccha and Kučera, 2017). It could, therefore, be proposed that the overall abundance of *G. truncatulinoides* in our subtropical settings was at least partly controlled by oceanic conditions occurring nearer to the sea surface (Mulitza et al., 1997; Jonkers and Kučera, 2016).

Sea surface water properties as well as vertical convective mixing in the Bahama region are closely related to the strength of

the atmospheric circulation as defined by the position of the ITCZ (e.g., Slowey and Curry, 1995; Wolff et al., 1999). Today, intensified trade winds coupled with cold meteorological fronts enhance upper water column mixing in the region through evaporative cooling during boreal winter, when the ITCZ is at the southernmost position (e.g., Wilson and Roberts, 1995). Previous studies from the western subtropical N. Atlantic have shown that time periods with reduced AMOC strength are consistent with southward displacements of the ITCZ and its associated rainfall belt, causing sea surface salinification (Schmidt

et al., 2006a; Carlson et al., 2008; Bahr et al., 2013). Acknowledging the fact that our study region lies too far north to be influenced by changes in the winter position of the ITCZ (Ziegler et al., 2008) – this would be of primary importance for modern-like winter-spring reproduction timing of *G. truncatulinoides* (Jonkers and Kučera, 2015) - we suggest that a southward displacement of the mean annual position of the ITCZ during T2 (Wang et al., 2004) could have promoted favorable conditions for *G. truncatulinoides* through generally strong sea surface cooling/salinification amplified by intensified

atmospheric circulation.

In the Bahamas, siliclastic input by other processes than wind transport is very limited, therefore, increased Fe content in the sediments could be attributed to enhanced trade winds strength which is coupled with a southern shift of the ITCZ (Roth and

Reijmer, 2004). Accordingly, the elevated XRF-derived Fe counts in our record during T2 (Fig. 5D) may support

intensification of trade winds and increased transport of Saharan dust at times of enhanced aridity over N. Africa, i.e., during

colder periods (Helmke et al., 2008; Tjallingii et al., 2008). Finally, increased velocities of the wind-driven Antilles Current

in the southwestern limb of the STG during glacial interval and T2 are thought to enhance winnowing of fine-grained material

on the northern slopes of LBB, which, together with the limited supply of aragonite needles, promoted enhanced sediment

consolidation (Chabaud et al., 2016).

### 6.3 Early MIS 5e

Various environmental properties (temperature, salinity, nutrients) can account for the proportional change in different

*Globigerinoides* species (Fig. 6). *G. sacculifer* – it makes up less than 5% of the planktic foraminiferal assemblage around the

LBB today (Siccha and Kučera, 2017) – is abundant in the Caribbean Sea and tropical Atlantic and commonly used as a tracer

of tropical waters and geographical shifts of the ITCZ (Poore et al., 2003; Vautravers et al., 2007). Also, *G. ruber* (pink) shows

rather coherent abundance maxima in the tropics (between 20°N and 20°S), while no such affinity is observed for *G. ruber*

(white) and *G. conglobatus* (Siccha and Kučera, 2017; Schiebel and Hemleben, 2017). Therefore, fluctuations in relative

abundances of *G. sacculifer* and *G. ruber* (pink) are referred here as to represent a warm tropical end-member (Fig. 1B).

As shown in Fig. 7B, relative abundances of the tropical species (here and further in the text *G. ruber* (pink) and *G. sacculifer*

calculated together) increased before the onset of the last interglacial "plateau" at ~129 ka. This transition was possibly coupled

with the intensification of the Gulf Stream at MIS 6/5e boundary (Bahr et al., 2011). In addition, a gradual rise in accumulation

of redox-sensitive element molybdenum (Mo) in sediment data from Cariaco Basin is observed across the penultimate

deglaciation (Fig. 7D). At that latter location, high Mo content is found in sediments deposited under anoxic conditions,

occurring only during warm interstadial periods associated with a northerly shifted ITCZ (Gibson and Peterson, 2014).

Accordingly, a gradual northward migration of the mean annual position of the ITCZ at the onset of MIS 5e could be implied.

In line with increasing low latitude summer insolation (Fig. 7C), this ITCZ displacement would also promote a northward

expansion of tropical pool waters (Ziegler et al., 2008). Because core MD99-2202 is located at the northern edge of the ITCZ

influence, the rapid shift in foraminiferal proportions at ~130 ka could, in fact, represent the onset of warm pool waters

influence, which resulted from a gradual northward-directed ITCZ movement. Similarly, the pronounced increase in the



tropical species relative abundances at 129-128 ka, a result of the rapid rise in *G. sacculifer* proportions (Figs. 6-7), may then

reflect a continuing northward expansion of the tropical pool waters. Abrupt sea surface warming at the onset of the interglacial

"plateau" is likewise found in some proxy records from the western subtropical N. Atlantic (Fig. 7E; Cortijo et al., 1999;

Deaney et al., 2017). Within some age uncertainties such a switch to warmer conditions could, however, correspond to the

rapid rise in accumulation of Mo in sediments from the Cariaco Basin (Fig. 7D).

Further, our data reveal a millennial-scale cooling/salinification event at ~127 ka, characterized by decreased proportions of

tropical foraminifera and elevated planktic $\delta^{18}O$ values (Figs. 5-7). That the abrupt cooling across the entire upper water

column occurred at the onset of the event is indicated by the re-occurrence of cold-water species *G. falconensis* and *G. inflata*

coincident with brief positive excursions in $\delta^{18}O$ of shallow and deep-dwelling foraminifera. A coherent cooling event, dated

by U-Th to be centered around 127 ka, is also evident in an isotopic record from the southwestern slope of the LBB (Slowey

et al., 1996; Henderson et al., 2000), suggesting at least a regional expression of the event. Simultaneously, the XRF record

from the Cariaco sediments reveals a stadial-like Mo-depleted (ITCZ southward) interval (Fig. 7D). The close similarity

between the tropical-species record from the Bahamas and the XRF data from Cariaco Basin supports the hypothesis that the

annual displacements of the ITCZ are also documented in our faunal counts. Moreover, because the aforementioned abrupt

climatic shift at ~127 ka cannot be reconciled with the insolation changes, additional forcings at play during early MIS 5e

should be considered.

Although the full resumption of the AMOC from a shallow or weak mode during T2 occurred only by ~124 ka, several studies

show that the AMOC abruptly recovered at the beginning of MIS 5e, apparently due to a deepened winter convection in the

Labrador Sea (Adkins et al., 1997; Galaasen et al., 2014; Deaney et al., 2017). In accordance with previous studies from the

tropical N. Atlantic suggesting a coupling between ITCZ position and ocean overturning (Rühlemann et al., 1999; Schmidt et

al., 2006a; Carlson et al., 2008), it could be argued that the northward ITCZ shift coeval with the rise in tropical foraminifera

proportions at our site at ~129-128 ka was consistent with the deepening of NADW and resumption of the AMOC (Fig. 7E).

In turn, the millennial-scale climatic reversal between 127 and 126 ka could have been related to the known reductions of deep

water ventilation during early MIS 5e (Galaasen et al, 2014; Deaney et al., 2017). A corresponding cooling and freshening

event – referred elsewhere as to a Younger Dryas type event - is captured in some high- and mid-latitude N. Atlantic records

(Bauch et al., 2012; Irvali et al., 2012; Schwab et al., 2013; Govin et al., 2014; Jiménez-Amat and Zahn, 2015; Zhuravleva et al., 2017a). Coherently with the Younger Dryas type cooling and the reduction/shallowing in the NADW, an increase in

Antarctic Bottom Water formation is revealed in the Southern Ocean core data, arguing for existence of an "interglacial" bipolar seesaw (Hayes et al., 2014). The out-of-phase climatic relationship between high northern and high southern latitudes, typical for the last glacial termination (Barker et al., 2009), could be attributed to a strong sensitivity of the transitional climatic regime of early MIS 5e due to persistent high-latitude freshening (continuing deglaciation) and suppressed overturning in the Nordic Seas. This is important because it helps to explain such a late occurrence of the Younger Dryas type event during T2,

when compared to the actual Younger Dryas in the last deglaciation.

**6.4 Late MIS 5e**

Relative abundances of tropical foraminifera in our core suggest an early SST maximum (between 128 and 124 ka) in the low-latitude N. Atlantic, which agrees in time with the recent compilation of global MIS 5e SST (Hoffmann et al., 2017). As the insolation forcing decreased during late MIS 5e and the ITCZ gradually moved southward (Fig. 7C-D), the white variety of

*G. ruber* started to dominate the assemblage (Fig. 6), arguing for generally colder sea surface conditions. The inferred broad salinity tolerance of this species, also to neritic conditions (Bé and Tolderlund, 1971; Schmuker and Schiebel, 2002), was used in some studies to link high proportions of *G. ruber* (pink and white varieties) with low surface salinities (Vautravers et al., 2007; Kandiano et al., 2012). The plots of the global distribution pattern of *G. ruber* (white) and *G. ruber* (pink), however, suggest that when relative abundances of these two species are approaching maximum values (40% and 10%, respectively),

the sea surface salinities would be higher for specimens of the white variety of *G. ruber* (Hilbrecht, 1996). Therefore, the strongly dominating white vs. pink *G. ruber* variety observed in our records during late MIS 5e could be linked not only to decreasing SST, but also to increasing sea surface salinity.

In their study from the western STG, Bahr et al. (2013) also reconstruct sea surface salinification during late MIS 5e in response to enhanced wind stress at times of deteriorating high-latitude climate and increasing meridional gradients. Accordingly, our

isotopic and faunal data (note the abrupt decrease in *G. sacculifer* proportion at 120 ka; Fig. 6B) suggest a pronounced climatic shift that could be attributed to a so-called "neoglaciation", consistent with the sea surface cooling in the western Nordic Seas and the Labrador Sea (Van Nieuwenhove et al., 2013; Irvali et al., 2016) as well as with a renewed growth of terrestrial ice

(Fronval and Jansen, 1997; Zhuravleva et al., 2017b).

Notably, a small but coherent increase in the aragonite content and Sr counts is evident at 120 ka and coincides with the change

towards finer-grained sediments, altogether arguing for a change in sedimentary regime before the end of the major flooding

interval at ~117 ka (Figs. 2 and 4). Further interpretations of the aragonite changes based on the available data appear rather

speculative, given that the aragonite precipitation on the platform top at times of sea level highstand is controlled by level of

aragonite supersaturation, which, in turn, depends on a number of climate-related parameters, such as $CO_2$ amounts,

temperature, salinity, water depth above the bank top as well as residence time of the water mass above the platform (Morse

and Mackenzie, 1990; Morse and He, 1993; Roth and Reijmer, 2004; 2005). Nevertheless, the coherent shift in the carbonate

minerology revealed after 120 ka may support major oceanographic and atmospheric changes during the late phase of MIS 5e

possibly coupled with a significant sea level change.

## 7 Conclusions

New isotopic, faunal and XRF evidence combined with published sedimentological data from a sediment core obtained from

the slope of the LBB were studied for changes in water masses, sedimentary regimes, and RSL change across the last

interglacial. By using new data, we were able to better constrain the last interglacial cycle in the investigated core section (cf.

Lantzsch et al., 2007). Elemental analyses, aragonite content and foraminiferal abundance records suggest that the LBB

became rapidly flooded at ~129 ka. The carbonate platform remained submerged until late MIS 5e, ca. 117 ka, implying a

prolonged interval with a RSL above -6 m. Although our data do not allow us to reconstruct the exact timing of the last

interglacial sea level peak, our sedimentological proxies point to changing sedimentary regimes on the slope of the LBB,

possibly as a result of intra-interglacial sea level and/or climatic fluctuations.

The overall climatic evolution of the last interglacial cycle in the Bahama region was closely coupled with ITCZ movements,

which, in turn, were the result of insolation and the AMOC forcing:

1. Termination 2: Strongly reduced $\delta^{18}O$ gradients between mixed-layer and thermocline-dwelling foraminifera suggest

decreased water column stratification. High proportions of *G. truncatulinoides* could be attributed to a deep vertical mixing

as a result of sea surface cooling/salinification and intensified trade winds strength at times of ITCZ being depressed far to

the south.

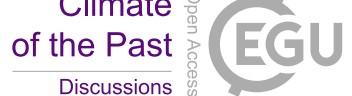

2. Early MIS 5e: Computed together, relative abundances of tropical foraminifera *G. sacculifer* and *G. ruber* (pink) agree well with the published ITCZ-related Cariaco record (Gibson and Peterson, 2014), suggesting climatic coupling between the regions. Based on these data, a northward displacement of the mean annual ITCZ position, in line with strong insolation forcing, could be inferred for early MIS 5e. However, an abrupt climatic shift intersected the early MIS 5e warmth. This so-called Younger Dryas type cooling event likely involved AMOC-related forcing that influenced (sub)tropical climate. The relatively late occurrence of Younger Dryas type cooling event, when compared to the actual Younger Dryas in the last deglaciation, is attributed to the transitional climatic regime of early MIS 5e, characterized by persistent high-latitude freshening and unstable deep-water overturning in the N. Atlantic.

3. Late MIS 5e: Overall sea surface cooling and possibly salinification is reconstructed for the Bahama region, in accordance with insolation decrease and a gradual southward displacement of the mean annual ITCZ. A coherent change is observed in faunal, isotopic and sedimentological proxies, arguing for coupled oceanic and northern hemisphere cryospheric reorganizations before the end of the major flooding period.

**Acknowledgments**

We wish to thank H. Lantzsch and J.J.G. Reijmer for providing us with the sediment core and data from core MD99-2202, S. Fessler for performing measurements on stable isotopes, S. Müller and D. Garbe-Schönberg for technical assistance during XRF scanning, J. Lübbers for her help with sample preparation, and E. Kandiano for introduction into tropical foraminiferal assemblages. A.Z. acknowledges funding from German Research Foundation (DFG grant BA1367/12-1).

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



**Figure 1: Maps showing positions of investigated core records and atmospheric/oceanic circulation.** (A) Simplified

surface water circulation in the (sub)tropical N. Atlantic and positions of investigated sediment records: MD99-2202 (27°34.5′



N, 78°57.9′ W, 460 m water depth; *this study*), Ocean Drilling Program (ODP) Site 1002 (10°42.7′ N, 65°10.2′ W, 893 m

water depth; Gibson and Peterson, 2014) and ODP Site 1063 (33°41.4′ N, 57°37.2′ W, 4584 m water depth; Deaney et al.,

2017). **(B)** Relative abundances of tropical foraminifera *G. sacculifer* and *G. ruber* (pink) (Siccha and Kučera, 2017) and

positions of the intertropical convergence zone (ITCZ) during boreal winter and summer. **(C)** Summer and winter hydrographic

sections (black line in **B)**, showing temperature and salinity obtained from World Ocean Atlas (Levitus, 2013). Vertical bars

denote calcification depths of *G. ruber* (white) and *G. truncatulinoides* (dex), respectively. Note, that today *G. truncatulinoides*

(dex) reproduce in winter time (Jonkers and Kučera, 2015) and due to its life cycle with changing habitats (as shown with

arrows) accumulate signals from different water depths. NEC – North Equatorial Current, AC – Antilles Current, FC – Florida

Current, STG – subtropical gyre. Maps are created using Ocean Data View (Schlitzer, 2016).





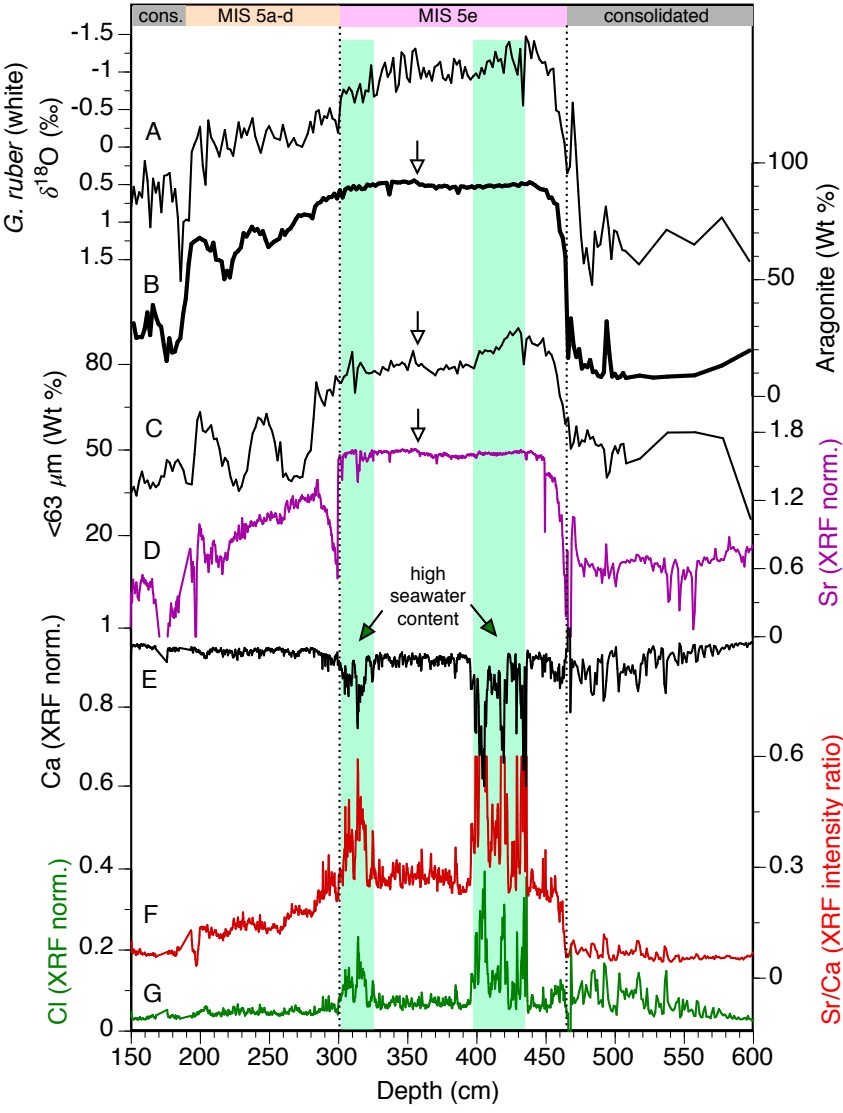

**Figure 2: XRF-scan results and sedimentological data from core MD99-2202.** (**A**) $\delta^{18}O$ values in *G. ruber* (white); (**B**)

aragonite content; (**C**) fraction with grain size <63 μm; (**A-C**) is from Lantzsch et al. (2007). Normalized elemental intensities

of (**D**) Sr, (**E**) Ca and (**G**) Cl and (**F**) Sr/Ca intensity ratio. Green bars denote core intervals with biased elemental intensities

due to inferred high seawater content (see main text). The white arrows mark a coherent change in sedimentological proxies

at 350 cm (**B-D**).





**Figure 3: Chronology of core MD99-2202.** Age model is based on alignment of (**D**) relative abundance record of *Globigerinoides* species and (**B**) planktic δ[18]O values (Lantzsch et al., 2007) with (**A**) global benthic isotope stack LS16 (Lisiecki and Stern, 2016). (**C**) Aragonite content (Lantzsch et al., 2007) and (**E**) relative abundances of *G. menardii* and *G. menardii* flexuosa are shown to support the stratigraphic subdivision of MIS 5.



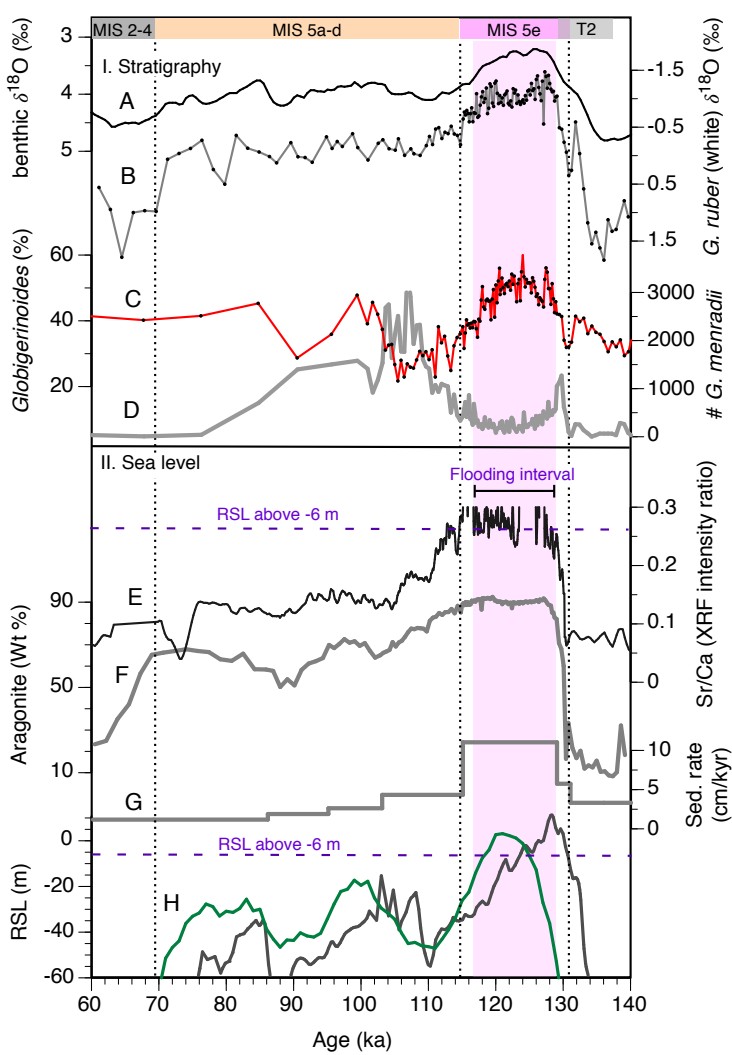

**Figure 4: Sedimentological and foraminiferal data from core MD99-2202**. Proxy records are compared to (**A**) global benthic isotope stack LS16 (Lisiecki and Stern, 2016) and (**H**) relative sea level (RSL) estimates (grey is from Grant et al. (2012); green is from Spratt and Lisiecki (2016)). (**B**) $\delta^{18}$O values in *G. ruber* (white) (Lantzsch et al., 2007), (**C**) relative abundances of *Globigerinoides* species, (**D**) absolute abundances of *G. menardii* per sample, (**E**) Sr/Ca intensity ratio, (**F**) aragonite content (Lantzsch et al., 2007), (**G**) computed sedimentation rates. Inferred platform flooding interval (lilac bar) is consistent with the enhanced production of Sr-rich aragonite needles and a RSL above -6 m (Chabaud et al., 2016). T2 – refers to the position of the penultimate deglaciation (Termination 2).





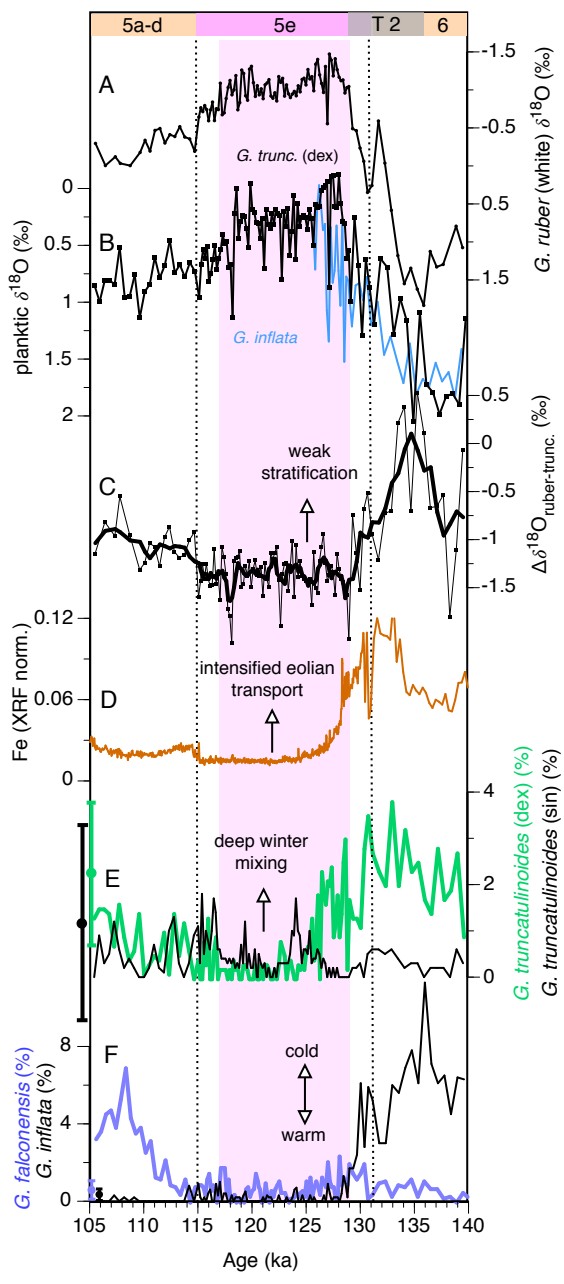

**Figure 5: Proxy records from core MD99-2202 over the last interglacial cycle.** (**A**) δ[18]O values in *G. ruber* (white) (Lantzsch et al., 2007), (**B**) δ[18]O values in *G. truncatulinoides* (dex) and *G. inflata,* (**C**) gradient Δδ[18]O between δ[18]O values in *G. ruber* (white) and *G. truncatulinoides* (dex), (**D**) normalized Fe intensities, (**E**) relative abundances of *G. truncatulinoides* (dex) (green) and *G. truncatulinoides* (sin) (black), (**F**) relative abundances of *G. falconensis* (violet) and *G. inflata* (black).





Also shown in (**E**) and (**F**) are modern relative foraminiferal abundances (average value ±1σ) around Bahama Bank, computed

using 7 nearest samples from Siccha and Kučera (2017) database. Shaded in lilac is the platform flooding interval (as defined

in Fig. 4). T2 – Termination 2.

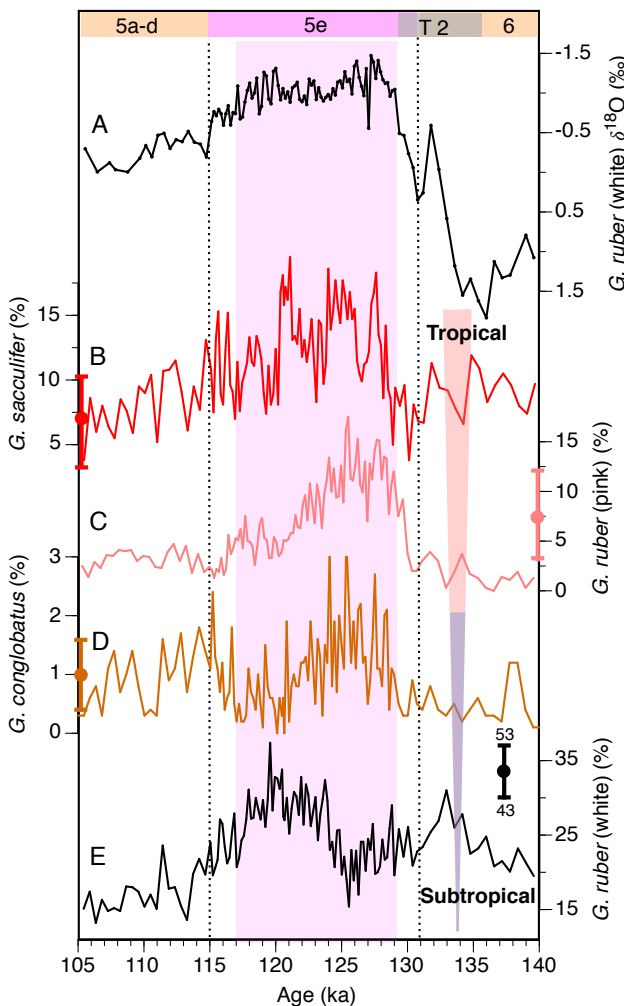

**Figure 6: Relative abundances of main *Globigerinoides* species in core MD99-2202.** (**A**) δ¹⁸O values in *G. ruber* (white)

(Lantzsch et al., 2007), relative abundances of (**B**) *G. sacculifer*, (**C**) *G. ruber* (pink), (**D**) *G. conglobatus*, (**E**) *G. ruber* (white).

Also shown in (**B-E**) are modern relative foraminiferal abundances (average value ±1σ) around Bahama Bank, computed using

7 nearest samples from Siccha and Kučera (2017) database. Shaded in lilac is the platform flooding interval (as defined in Fig.

4). T2 – Termination 2.



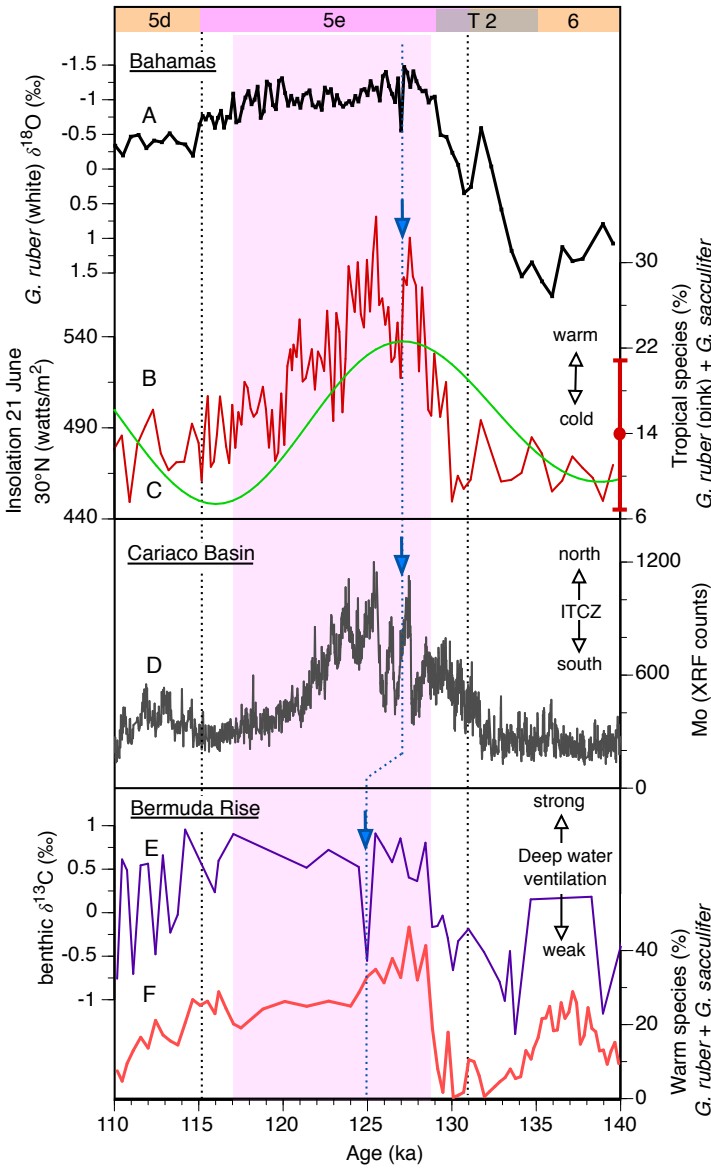

**Figure 7: Comparison of proxy records from (sub)tropical N. Atlantic. (A)** $\delta^{18}O$ values in *G. ruber* (white) in core MD99-

2202 (Lantzsch et al., 2007), **(B)** relative abundances of tropical species *G. sacculifer* and *G. ruber* (pink) in core MD99-2202,

**(C)** boreal summer insolation (21 June, 30° N), computed with AnalySeries 2.0.8 (Paillard et al., 1996) using Laskar et al.

(2004) data, **(D)** molybdenum (Mo) record from ODP Site 1002 (Gibson and Peterson, 2014), **(E-F)** $\delta^{13}C$ values measured in

benthic foraminifera and relative abundances of *G. ruber* (total) and *G. sacculifer* from ODP Site 1063 (Deaney et al., 2017).



Also shown in (**B**) are modern relative abundances of *G. sacculifer* and *G. ruber* (pink) (average value ±1σ) around Bahama

Bank, computed using 7 nearest samples from Siccha and Kučera (2017) database. The blue arrows and the dashed line suggest

correlation of events (so-called Younger Dryas type cooling) in the subtropical and tropical N. Atlantic (see text). Shaded in

lilac is the platform flooding interval (as defined in Fig. 4). T2 – Termination 2.