# Peer review of "Last interglacial ocean changes in the Bahamas: climate"

_Climate of the Past, 2018_

## Referee Comment (RC1) · A. Bahr (Referee) · 30 Apr 2018

GENERAL REMARKS: The authors present a comprehensive collection of faunal, stable isotope and sediment-geochemical data from Little Bahama Bank (LBB) core MD99-2202 encompassing MIS 5e in high temporal resolution. Such high-resolution low-latitude (27°N) records of the penultimate Interglacial are rare, but important to constrain the climatic variability of previous interglacials when compared to the Holocene. The authors argue that the surface ocean variability at LBB reflects changes in the position of the subtropical gyre and tropical warm pool, responding to latitudinal shifts of the ITCZ that are driven by insolation and AMOC changes. In addition, the sea level

history at LBB is discussed, mainly based on the sedimentary composition (aragonite content) of the sediment. In general, the author's interpretations are well-founded by proxy evidence and supported by previous studies. Some problematic aspects of the interpretation are discussed below, but do not interfere with the general messages of the paper.

The manuscript is generally well-written and the study undoubtedly has its merits as a valuable contribution for the understanding of low-latitude climate variability during MIS 5e as well as the low-high-latitude feedbacks. However, the manuscript lacks a clear focus. This regards in particular the introduction -it should include more concise statements regarding the aims of the study, e.g. hypotheses to be tested and specific questions that should be solved. At the moment the introductory paragraphs (as well as the abstract ad conclusions) are very general, partly with a focus that hinges strongly on local aspects of sedimentary dynamics at LBB. Hence, I would strongly advocate to sharpen the focus of the manuscript, as the reader is left with the impression that the study confirms previous conceptual models (e.g. regarding the displacement of the ITCZ during MIS 5e) but wonders about the specific take-home-messages and new insights retrieved from this study. I therefore recommend the authors to re-write the respective parts of their manuscript (in particular the introduction; see also specific comments below) to avoid underselling of their data.

SPECIFIC COMMENTS:

Abstract: As discussed above, the abstract should be more specific about what exactly the authors want to study. At present, the first three sentences concentrate on the local/regional aspects concerning LBB, but in fact the data can be used to infer much more general insights into low-high-latitude feedbacks and subtropical gyre dynamics and Gulf Stream variability. Hence, I suggest to reduce the reference to LBB but focus on the broader context.

Introduction: line 40 and elsewhere: I would avoid abbreviating North Atlantic as "N.

Atlantic" l. 50: "…. we attempt to close this gap…" reflects the problem of the introduction - this is far too general. Data generation per se is important, but should be done with some hypothesis/question to be tackled in mind. At the moment I also miss a more specific lay out of the controversies that are mentioned. This would help to formulate specific questions and hypotheses at the end of the introduction.

Regional setting: l. 64: capitalize "intertropical convergence zone" l. 71: replace "tropical pool waters" with "tropical warm pool" l. 73: "thermocline layer" is too unspecific. Does this refer to the permanent thermocline?

Methods: There should be a short statement in the introduction about the type of proxies used. In the present state, the purpose of the different proxies is unclear until the discussion. However, I would expect to read one or two sentences about the rational for XRF scanning, why $\delta$18O of deep and shallow dwellers were used, and about the purpose of the faunal studies. Again, the mentioning of the proxies can be done in conjunction with the layout of the specific goals in the introduction (see comments above).

Results: l. 163: "physical sediment properties" should be replaced by "sedimentological properties", this seems more appropriate as it refers to the grain size curve shown. l. 174: I agree that the "significant sedimentological shift" mentioned here is no artifact as it displays in different, independent proxies. However, given its minute amplitude relative to the general fluctuations in the core it is an overstatement to call it "significant". Considering the rather diffuse discussion in Section 6.4 I would skip the reference and discussion of this feature (see also respective comment below). l. 178: "during the major deglacial transition … low isotopic gradients…" this statement does not fully reflect the data, as there is a steady trend to more stratification from 135-129 ka, reaching the MIS 5e level of well-stratified waters. As written in the text it sounds like the entire transition is characterized by a persistent low isotopic gradients. l. 181: please call out Fig. 6 after "species are observed" l. 185: please call out Fig. 5 after "abruptly increase". Also note that the variations of G. trunca (sin) in Fig. 5e are within

the 1-sigma error of their present-day abundances. Is it necessary to plot these G. trunca (sin) abundances?

Discussion: l. 192-211: I wonder about the necessity to discuss the Sr/Ca record. In principle this is a good proxy for aragonite, however, the authors make the convincing case that this record is biased by changes in porosity and water content. Considering that the authors present the XRD-based record of aragonite form Lantzsch et al. (2007), the discussion of Sr/Ca can be omitted without losing information. l. 222: Please add a reference for the subsidence rate of the LBB l. 228: if I am correct, the sea level rise should be between 12-15 m (15 = 9 + 6 m) not 12-16 m. Please check. l. 256: "warm/cold conditions" – please specify what is meant here. l. 270-272: In principle I agree with the interpretation that G. truncatulinoides abundances strongly depend on the upper ocean stratification. However, in this respect, it is interesting that G. trunca. (dex) is still high during late MIS 5e, when ïĄĎ$\delta$18O is already low. Hence, vertical water column stratification is not the sole factor influencing the G. trunc. abundances. l. 283-287: Fe appears to lag ïĄĎ$\delta$18O, hence, question is if dust is really the dominant factor that governs the Fe abundances if ïĄĎ$\delta$18O is supposed to be the prime proxy recording for wind-driven water column homogenization. Fe might also be influenced by diagenetic processes, hence, it would be worthwhile looking at Ti/Al as Ti is not influenced by diagenesis. l. 287-291: to check if winnowing plays a role during the deglaciation elemental ratios such as Zr/Rb or Zr/Al might be used to check for high bottom current velocities (Bahr et al., 2014). l. 332: please add a reference after "only by ∼124 ka" l. 355: correct for "Hofman et al." (not Hofmann) l. 364: Notably, the ruber (w) abundances are strikingly similar to the $\delta$18Oivf-sw record of G. ruber (w) from ODP Site 1058 (Bahr et al, 2013). This supports the view that salinity is the main driver of G. ruber (w) abundances. l. 372-381: as mentioned above, the discussed changes in Sr and aragonite content are really minute compared to the other variations observed in the proxy records. Given that the authors make only very general inferences about the paleoclimatic implications I suggest to remove this paragraph.

Conclusions l. 387: "in the investigated core section": this is much too local, especially for the conclusions (see also my general comments). The broader implications of this study should become clear here. l. 389-392: these statements regarding the local sedimentological processes on LBB are quite general and not novel considering the amount of publications dealing with this topic. l. 398: replace "depressed" by "shifted"

Figures: Fig. 4A-C is repetitive of Fig. 3 Fig. 4E: if Sr/Ca remains in the figure (see comments above): this record has been truncated at 0.3, please state this in the captions. Fig. 7E-F: Is it necessary to show G. ruber and G. sacculifer abundances from ODP Site 1063 here?

Reference: Bahr, A., Jiménez-Espejo, F.J., Kolasinac, N., Grunert, P., Hernández-Molina, F.J., Röhl, U., Voelker, A.H., Escutia, C., Stow, D.A. and Hodell, D. (2014) Deciphering bottom current velocity and paleoclimate signals from contourite deposits in the Gulf of Cádiz during the last 140 kyr: An inorganic geochemical approach. Geochemistry, Geophysics, Geosystems, 15, 3145-3160.

---

## Referee Comment (RC2) · Anonymous Referee #2 · 17 May 2018

SUMMARY: Zhuravleva and Bauch present a detailed consideration of the climate evolution of the Last Interglacial (LIg) for a core site on the Little Bahama Bank (LBB) using faunal assemblage and scanning XRF techniques. The high resolution faunal assemblages nicely resolve hydrographic oscillations at the site for the LIg reflecting both the insolation driven and AMOC modulated migration of the ITCZ for this region.

I would recommend the following amendments/clarifications: (a) change of title to better reflect the content of the paper; (b) removal or at the very least restructuring of the discussion of sea level. This section could be significantly trimmed and simplified (no new insights offered but a nice corroboration). Alternatively, if the authors wish to retain

the sea-level discussion, then discussion of other sea level evidence from the region, glacio-isostatic adjustment (GIA) processes etc. are needed. (c) clearer discussion of the teleconnections between N Atlantic oceanic changes (i.e., variation in AMOC), the migration of the ITCZ and surface hydrographic change at MD99-2202.

GENERAL COMMENTS: The manuscript, in general, reads well. However, the structure and focus of the paper requires further thought. A clear statement of the research questions was missing and is reflected in the general tone of the introduction (and the manuscript generally).

1. Title

I found this to be somewhat misleading. The data in Zhuravleva and Bauch is not a sea-level record per se, rather a record of increased aragonite supply to the core site during interglacials, with these intervals of increased aragonite production/supply likely corresponding to < -6 m relative sea level (RSL) due to the generally shallow nature of Little Bahama Bank (i.e., you can infer periods of <-6 m relative sea level). This work nicely corroborates the Lantzsch et al., 2007 and Chabaud et al., 2016 studies but isn't a sea-level story. What is new and interesting the palaeoceanographic evolution of the Last Interglacial (LIg) at the site, and the interplay of interglacial climate (movement of the ITCZ etc.). I would suggest changing the title to better reflect this.

2. Sea level

This section requires some restructuring to help the reader. The definition of the "flooding interval" (and corresponding relative sea level, <-6 m) is key to this section of the manuscript but I struggled to clearly follow the logic of how you defined the flooding interval using your records and why a -6 m RSL for this interval was appropriate. The connection between the flooding interval and inferred RSL of < -6 m was found almost at the end of the section (line 222 to 226) when it should be at the start. All the information is there but the reader has to work hard to follow the argument.

Perhaps something along the lines of;

modern LBB lagoon is shallow, with an average water depth < 6 m (Williams, 1985); tectonic stability of the region (refs needed);

during the LIg, increasing RSL at the site floods the generally shallow bank and increases the area for aragonite production (i.e., the carbonate shedding model, Droxler and Schlager, 1985; Schlager et al., 1994);

Conversely, during glacial intervals, the top is exposed which limits the production and export of aragonite;

As such, we define the flooding interval (and inferred <-6 m RSL) is defined by an increase in the sedimentation rate, increase in wt % aragonite, increased Sr/Ca ratio, increase % Globigerinoides/decrease in numbers of G. menardii.

This could then usefully be followed with your discussion of very high values of Sr/Ca due to increased saltwater (lines 192 to 211). Perhaps shade these 'problematic' Sr/Ca intervals in subsequent figures? You should also note the truncation of the Sr/Ca record in caption of Figure 4.

I would suggest confining discussion of sea level in this section to that suggested above. If you wish to make more of the sea level story, then greater consideration of other Bahamas sea-level records, as well as those from the wider area is needed. For example, the +6m notch on Little Sale Cay (LLB) (Neumann and Hearty, 1996), other geomorphological records (e.g., Hearty and Kindler, 1995; Neumann and Hearty, 1996), the elevated Last Interglacial (LIg) coral records of Chen et al 1991, Hearty et al., 2007, Thompson et al., 2011 and the regionally extensive erosional surface that is suggestive of a sea-level oscillation within the LIg (Bahamas, Florida and Yucatan; Chen et al 1991, Hearty et al., 2007, Blanchon et al., 2009; Thompson et al., 2011).

How does the timing of the highstand from the coral/other records from the Bahamas compare to the timing of the interval of enhanced aragonite production (and inferred

sea levels < -6 m)?

How do changes in hydrography (variations in faunal assemblages) at the site compare to the timing of the Bahamas LIg highstand? The broad correspondence between climate ($\delta$18OG.ruber) and relative sea level (RSL) (weight % aragonite) is hinted at in lines 138 to 141 but could be developed further if you wish to keep the sea-level discussion.

Any discussion of the LIg highstand in a general sense (i.e., the eustatic record) (lines 227 to 231) and Bahamas RSL will need to consider glacio-isostatic (GIA) processes, given the intermediate location of the site on the peripheral bulge of the former Laurentide Ice Sheets. There will be a regional expression of the LIg highstand; the Bahamas would "see" a "late" LIg highstand compared to eustatic sea level (e.g., Figure 6 in Stirling et al., 1998). There seems to be good correspondence between the age of your "flooding interval" at the site (i.e., sea level < -6 m) and the predictions of RSL (Stirling et al., 1998, their Figure 6).

Given that the records presented are not strictly a sea level record, rather incidence of increased aragonite production/export, and seems to corroborate previous studies rather than adding anything new, I would confine this section to just a brief consideration of the timing of your "flooding interval".

3. Palaeoceanographic reconstruction

This section is much more coherent and well written. I would recommend this as the focus of the manuscript.

The discussion, while nicely documenting the site/regional changes during the LIG, was lacking in consideration of the mechanisms. This section would be strengthened by a clearer exposition of the mechanisms linking ITCZ position, insolation (precession and the migration of the ITCZ to the warming hemisphere) and AMOC (i.e., modification of the thermal condition at the surface, due to interactions with the ocean, that in

turn act to drive atmospheric circulation). This is well documented for the last deglaciation and glacial period (and in modelling studies), where N. hemisphere extra-tropical cooling (brought about by variations in the AMOC, forced by freshwater inputs) lead to an interhemispheric thermal gradient and a southward shift in the ITCZ (e.g., review of Chiang and Friedman, 2012 or Schneider et al., 2014). This would help the reader to place the different records (Cariaco, MD99-2202 and Site 1063) within a broader climatological context. Again, all the 'threads' of the story are there, it just needs a stranger framework. For example, I found the correspondence between the % G. ruber and G. sacculifer and the XRF Mo count of the Caricao Basin striking (demonstrating the clear record of ITCZ shifts at the LBB) but the link to N. Atlantic surface density changes (AMOC slowdown with surface freshening e.g., Galaasen et al., 2014 etc.) and positive the $\delta$18OG.ruber excursion and faunal changes at MD99-2202 and southward migration of the ITCZ (Cariaco Mo, MD99-2202 decrease in % Globigerinoides) weak. A short introductory paragraph should fix this.

A southward shift of the ITCZ, strengthens of the trade winds increases the eolian input from the Sahara, resulting in reduced Al/Ti in Cariaco. These episodes of decreased Al/Ti ratios in Cariaco correspond to elevated salinities in the Caribbean (e.g., Yarincik et al 2000). I assume there is clear correspondence between the Cariaco Al/Ti and Mo records and hence to your % of tropical species? Do you see an increase in iron (with increased dust transport) in your record during the positive the $\delta$18OG.ruber excursion $\sim$ 127 ka? (plotting this on a log scale for the LIg might help). Dust inputs are probably better reflected in the XRF core scanning Ti record, given that your Fe inputs could change with a number of factors. It would be interesting to compare to your faunal assemblages and a calculated $\delta$18Oseawater for MD99-2202. Additionally, is there any correspondence to the dated palaeosols on the Bahamas (Muhs et al., 2007)?

Given you have faunal assemblage data, could you calculate a transfer function/MAT sea surface temperature? From this you could then calculate $\delta$18Oseawater at the site to think about density changes during the LIg.

lines 372 to 381 – I found this paragraph to be highly speculative and not well supported by your data (I struggled to see the change in the sedimentological properties in your figures, even with the help of the white arrows). I would suggest removing this section.

TECHNICAL CORRECTIONS

1. Referencing: greater care needed with referencing. Please check manuscript. For example, the depth of submersion of the LBB (and origin of the -6 m quoted often in the paper) is Neumann and Land, 1975. Similarly the carbonate shedding model (upon which the inferred sea-level story is based) comes from Droxler and Schlager, 1985; Schlager et al., 1994

line 124 – what potential biases are you refereeing to? Please give appropriate references for these.

line 242 – what is the derivation of "minimal ice volume interval" and it's reference?

line 332 - reference required for the "full resumption of the AMOC. . . only by ∼124 ka"

2. Other: line 64 – capitalisation of "intertropical convergence zone"

line 133 – unit of measurement missing, add "m".

line 330 - please clarify or add examples of the "additional forcing" or add "as discussed below".

line 324-325 - Clarification of the age of the cooling/increase salinity event ∼ 127 ka is needed. The ∼127 ka age for this event is derived from your age model, whereas the U-series ages for the correlative event in core 152JPC (Bahamas, Slowey et al., 1996), dated (in duplicate) above and below the event to ∼121 ± 3 ka and 125.6 ka (mean of 127 ± 4.8 and 124.1 ± 5.1) respectively. (Note, the relatively large age uncertainties associated with these U-series ages)

3. Figures: typo - Axis label in Figure 4D should read "# G. menardii" rather than "# G. menradii" I would remove the sea level records (H), or if you choose to retain these and

your discussion of sea level, then you should remove the dashed blue line "RSL above -6 m" from the lowermost panel (H).

REFERENCES CITED: Blanchon, P., Eisenhauer, A., Fietzke, J. and Liebetrau, V., 2009. Rapid sea-level rise and reef back-stepping at the close of the last interglacial highstand. Nature, 458, 881.

Chabaud, L., Ducassou, E., Tournadour, E., Mulder, T., Reijmer, J. J. G., Conesa, G., Giraudeau, J., Hanquiez, V., Borgomano, J. and Ross, L.: Sedimentary processes determining the modern carbonate periplatform drift of Little Bahama Bank, Mar. Geol., 378, 213–229.

Chen, J.H., Curran, H.A., White, B., Wasserburg, G.J. 1991. Precise chronology of the last interglacial period: 234U-230Th data from fossil coral reefs in the Bahamas. Geological Society of America Bulletin 103, 82-97.

Chiang, J. C. H. & Friedman, A. R. Extratropical cooling, interhemispheric thermal gradients, and tropical climate change. Annu. Rev. Earth Planet. Sci. 40, 383–412 (2012)

Diz et al 2018. Ocean and atmosphere teleconnections modulate east tropical Pacific productivity at late to middle Pleistocene terminations. Erath and Planetary Science Letters 49, 82-91.

Droxler, A.W., Schlager, W., 1985. Glacial versus interglacial sedimentation rates and tur- bidite frequency in the Bahamas. Geology 13, 799–802.

Galaasen, E. V., Ninnemann, U. S., IrvalÄś, N., Kleiven, H. (Kikki) F., Rosenthal, Y., Kissel, C. and Hodell, D. A.: Rapid Reductions in North Atlantic Deep Water During the Peak of the Last Interglacial Period, Science, 343, 1129.

Hearty PJ, Kindler P. Sea-level highstand chronology from stable carbonate platforms (Bermuda and the Bahamas). Journal of Coastal Research. 1995 Jul 1:675-89.

Hearty, P.J., Hollin, J.T., Neumann, A.C., O'Leary, M.J. and McCulloch, M., 2007. Global sea-level fluctuations during the Last Interglaciation (MIS 5e). Quaternary Science Reviews, 26, 2090-2112.

Lantzsch, H., Roth, S., Reijmer, J. J. G. and Kinkel, H.: Seaâ̧level related resedimentation processes on the northern slope of Little Bahama Bank (Middle Pleistocene to Holocene), Sedimentology, 54, 1307–1322.

Mahowald, N. M., D. R. Muhs, S. Levis, P. J. Rasch, M. Yoshioka, C. S. Zender, and C. Luo (2006), Change in atmospheric mineral aerosols in response to climate: Last glacial period, preindustrial, modern, and doubled carbon dioxide climates, J. Geophys. Res., 111, D10202, doi:10.1029/2005JD006653.

Muhs, D.R., Budahn, J.R., Prospero, J.M., Carey, S.N. 2007. Geochemical evidence for African dust inputs to soils of western Atlantic islands: Barbados, the Bahamas, and Florida. Journal of Geophysical Research 112, F02009 doi:10.1029/2005JF000445.

Neumann, C.A. & Hearty PJ. 1996. Rapid sea-level changes at the close of the last interglacial (substage 5e) recorded in Bahamian island geology. Geology, 24(9):775-8.

Neumann, A.C., Land, L., 1975. Lime mud deposition and calcareous algae in the bight of Abaco, Bahamas: a budget. J. Sediment. Petrol. 45, 763–786.

Schlager, W., Reijmer, J.J.G., Droxler, A.W., 1994. Highstand shedding of carbonate plat- forms. J. Sediment. Res. B64, 270–281.

Schneider, T., Bischoff, T. & Haug, G. H. Migrations and dynamics of the intertropical convergence zone. Nature 513, 45–53 (2014).

Slowey, N.C., Henderson, G.M., Curry, W.B. 1996. Direct U–Th dating of marine sediments from the two most recent interglacial periods. Nature 382, 242-244.

Stirling, C.H., Esat, T.M., Lambeck, K., McCulloch, M.T. 1998. Timing and duration of the Last Interglacial: evidence for a restricted interval of widespread coral reef growth.

Earth and Planetary Science Letters 160, 745–762.

Thompson, W.G., Curran, H.A., Wilson, M.A. and White, B., 2011. Sea-level oscillations during the last interglacial highstand recorded by Bahamas corals. Nature Geoscience, 4, 684-687.

Williams, S.C., 1985. Stratigraphy, Facies Evolution and Diagenesis of Late Cenozoic Lime- stones and Dolomites, Little Bahama Bank, Bahamas. Univ. Miami, Coral Gables FL (217 pp.).

Yarincik, K.M., Murray, R.W. and Peterson, L.C., 2000. Climatically sensitive eolian and hemipelagic deposition in the Cariaco Basin, Venezuela, over the past 578,000 years: Results from Al/Ti and K/Al. Paleoceanography, 15(2), 210-228.
* * *
**CPD**

---

## Author Comment (AC1) · 12 Jul 2018

A. Bahr (Referee)

andre.bahr@geow.uni-heidelberg.de

Reviewer's comment: GENERAL REMARKS: The authors present a comprehensive collection of faunal, stable isotope and sediment-geochemical data from Little Bahama Bank (LBB) core MD99-2202 encompassing MIS 5e in high temporal resolution. Such high-resolution low-latitude (27°N) records of the penultimate Interglacial are rare, but

important to constrain the climatic variability of previous interglacials when compared to the Holocene.

The authors argue that the surface ocean variability at LBB reflects changes in the position of the subtropical gyre and tropical warm pool, responding to latitudinal shifts of the ITCZ that are driven by insolation and AMOC changes. In addition, the sea level history at LBB is discussed, mainly based on the sedimentary composition (aragonite content) of the sediment. In general, the author's interpretations are well-founded by proxy evidence and supported by previous studies. Some problematic aspects of the interpretation are discussed below, but do not interfere with the general messages of the paper.

The manuscript is generally well-written and the study undoubtedly has its merits as a valuable contribution for the understanding of low-latitude climate variability during MIS 5e as well as the low-high-latitude feedbacks. However, the manuscript lacks a clear focus. This regards in particular the introduction - it should include more concise statements regarding the aims of the study, e.g. hypotheses to be tested and specific questions that should be solved. At the moment the introductory paragraphs (as well as the abstract ad conclusions) are very general, partly with a focus that hinges strongly on local aspects of sedimentary dynamics at LBB. Hence, I would strongly advocate to sharpen the focus of the manuscript, as the reader is left with the impression that the study confirms previous conceptual models (e.g. regarding the displacement of the ITCZ during MIS 5e) but wonders about the specific take-home-messages and new insights retrieved from this study. I therefore recommend the authors to re-write the respective parts of their manuscript (in particular the introduction; see also specific comments below) to avoid underselling of their data.

Author's response: Please note that in accordance with the journal requirements, all changes in the manuscript are provided in a marked-up version (track changes in Word, converted into a *.pdf file). The line numbers used in the current Author's response refer to the marked-up version, which could be find in the supplement.

The introduction has been rewritten, in attempt to provide a clearer focus of the study. Aims, methods and problematics are now also discussed.

Reviewer's comment: SPECIFIC COMMENTS: Abstract: As discussed above, the abstract should be more specific about what exactly the authors want to study. At present, the first three sentences concentrate on the local/regional aspects concerning LBB, but in fact the data can be used to infer much more general insights into low-high-latitude feedbacks and subtropical gyre dynamics and Gulf Stream variability. Hence, I suggest to reduce the reference to LBB but focus on the broader context.

Author's response: The rewritten abstract focuses on millennial-scale teleconnections between high and low latitudes during the last interglacial. In addition, the strong freshening in the high latitudes during early MIS 5e is described as the main reason for the warm-cold switches, observed across various oceanic basins.

Reviewer's comment: Introduction: line 40 and elsewhere: I would avoid abbreviating North Atlantic as "N. Atlantic"

Author's response: Done.

Reviewer's comment: l. 50: ". . .. we attempt to close this gap. . ." reflects the problem of the introduction - this is far too general. Data generation per se is important, but should be done with some hypothesis/question to be tackled in mind. At the moment I also miss a more specific lay out of the controversies that are mentioned. This would help to formulate specific questions and hypotheses at the end of the introduction.

Author's response: The introduction has been rewritten (see the comment above).

Reviewer's comment: Regional setting: l. 64: capitalize "intertropical convergence zone"

Author's response: Done (ls. 15-16, 69-70, 1651).

Reviewer's comment: l. 71: replace "tropical pool waters" with "tropical warm pool"

Author's response: Replaced with "the Atlantic Pool Water" (ls. 117, 234).

Reviewer's comment: l. 73: "thermocline layer" is too unspecific. Does this refer to the permanent thermocline?

Author's response: Changed to the permanent thermocline (l. 237).

Reviewer's comment: Methods: There should be a short statement in the introduction about the type of proxies used. In the present state, the purpose of the different proxies is unclear until the discussion. However, I would expect to read one or two sentences about the rational for XRF scanning, why $\delta$18O of deep and shallow dwellers were used, and about the purpose of the faunal studies. Again, the mentioning of the proxies can be done in conjunction with the layout of the specific goals in the introduction (see comments above). Author's response: The used proxies are mentioned in the rewritten introduction together with their specific goals (ls. 119-128).

Reviewer's comment: Results: l. 163: "physical sediment properties" should be replaced by "sedimentological properties", this seems more appropriate as it refers to the grain size curve shown.

Author's response: The text has been rephrased. Now "sedimentological records" are used (l. 389).

Reviewer's comment: l. 174: I agree that the "significant sedimentological shift" mentioned here is no artifact as it displays in different, independent proxies. However, given its minute amplitude relative to the general fluctuations in the core it is an overstatement to call it "significant". Considering the rather diffuse discussion in Section 6.4 I would skip the reference and discussion of this feature (see also respective comment below).

Author's response: This section has been deleted from the results as well as from the discussion.

Reviewer's comment: l. 178: "during the major deglacial transition . . . low isotopic

gradients. . ." this statement does not fully reflect the data, as there is a steady trend to more stratification from 135-129 ka, reaching the MIS 5e level of well-stratified waters. As written in the text it sounds like the entire transition is characterized by a persistent low isotopic gradients.

Author's response: We fully agree with the Reviewer's comment. The statement has been changed to "During the penultimate glacial maximum <. . .> gradients <. . .> are very low, succeeded by a gradually increasing difference across the T2, ∼135-129 ka" (ls. 425-427). Reviewer's comment: l. 181: please call out Fig. 6 after "species are observed" Author's response: Done (l. 431).

Reviewer's comment: l. 185: please call out Fig. 5 after "abruptly increase". Also note that the variations of G. trunca (sin) in Fig. 5e are within the 1-sigma error of their present-day abundances. Is it necessary to plot these G. trunca (sin) abundances?

Author's response: Fig. 5 (now Fig. 4) is called out after "together with a reappearance of G. inflata" (ls. 433). For simplification and clarity, relative abundances of G. truncatulinoides (sin) as well as G. falconensis are not shown in figures any more.

Reviewer's comment: Discussion: l. 192-211: I wonder about the necessity to discuss the Sr/Ca record. In principle this is a good proxy for aragonite, however, the authors make the convincing case that this record is biased by changes in porosity and water content. Considering that the authors present the XRD-based record of aragonite form Lantzsch et al. (2007), the discussion of Sr/Ca can be omitted without losing information.

Author's response: The discussion of our Sr/Ca record, in particular, of the "problematic intervals" is retained, however, has been substantially shortened (ls. 440-544).

Reviewer's comment: l. 222: Please add a reference for the subsidence rate of the LBB

Author's response: Done (study by Carew and Mylroie (1995) is cited, l. 441)

Reviewer's comment: l. 228: if I am correct, the sea level rise should be between 12-15 m (15 = 9 + 6 m) not 12-16 m. Please check.

Author's response: Correct. This paragraph has been, however, deleted.

Reviewer's comment: l. 256: "warm/cold conditions" - please specify what is meant here.

Author's response: Has been changed to "temperature estimations during late MIS 6 so far reveal cold subsurface conditions" (l. 670).

Reviewer's comment: l. 270-272: In principle I agree with the interpretation that G. truncatulinoides abundances strongly depend on the upper ocean stratification. However, in this respect, it is interesting that G. trunca. (dex) is still high during late MIS 5e, when $\delta$18O is already low. Hence, vertical water column stratification is not the sole factor influencing the G. trunc. abundances.

Author's response: We agree with the Reviewer's comment. A paragraph, dealing with additional forcing factors controlling occurrences of G. truncatulinoides (dex) and G. inflata has been included (ls. 688-803).

Reviewer's comment: l. 283-287: Fe appears to lag $\delta$18O, hence, question is if dust is really the dominant factor that governs the Fe abundances if $\delta$18O is supposed to be the prime proxy recording for wind-driven water column homogenization. Fe might also be influenced by diagenetic processes, hence, it would be worthwhile looking at Ti/Al as Ti is not influenced by diagenesis.

Author's response: We have reconsidered our Fe data and has significantly restricted the interpretation, given the "variety of additional effects that may have influenced our Fe-record" (ls. 816-820). Ti content in the investigated sediment core appears to be very low, in addition, possibly strongly affected by seawater content (Fig. 1).

Reviewer's comment: l. 287-291: to check if winnowing plays a role during the deglaciation elemental ratios such as Zr/Rb or Zr/Al might be used to check for high

bottom current velocities (Bahr et al., 2014).

Author's response: Increased winnowing at the northern slope of Little Bahama Bank during glacial times (a result of an intensified wind-driven Antilles Current) was previously suggested by Chabaud et al. (2016). As we could not prove the statement by using Zr/Rb or Zr/Al data (Fig. 1), as suggested above, this part has been removed.

Reviewer's comment: l. 332: please add a reference after "only by ~124 ka"

Author's response: Done (l. 1081-1082).

Reviewer's comment: l. 355: correct for "Hofman et al." (not Hofmann)

Author's response: Done. Changed for Hoffman et al.

Reviewer's comment: l. 364: Notably, the ruber (w) abundances are strikingly similar to the $\delta$18Oivf-sw record of G. ruber (w) from ODP Site 1058 (Bahr et al, 2013). This supports the view that salinity is the main driver of G. ruber (w) abundances.

Author's response: We agree with the comment above, however, a common temporal framework is needed for better core-to-core correlation and further climatic implications (Fig. 2).

Reviewer's comment: l. 372-381: as mentioned above, the discussed changes in Sr and aragonite content are really minute compared to the other variations observed in the proxy records. Given that the authors make only very general inferences about the paleoclimatic implications I suggest to remove this paragraph.

Author's response: Done. The paragraph has been removed.

Reviewer's comment: Conclusions l. 387: "in the investigated core section": this is much too local, especially for the conclusions (see also my general comments). The broader implications of this study should become clear here.

Author's response: Conclusions have been rewritten to emphasize the broader impli-

cations of the study (particularly, the third paragraph).

Reviewer's comment: l. 389-392: these statements regarding the local sedimentological processes on LBB are quite general and not novel considering the amount of publications dealing with this topic.

Author's response: The statements have been removed from the conclusions.

Reviewer's comment: l. 398: replace "depressed" by "shifted"

Author's response: Done (l. 1191).

Reviewer's comment: Figures: Fig. 4A-C is repetitive of Fig. 3 Fig. 4E: if Sr/Ca remains in the figure (see comments above): this record has been truncated at 0.3, please state this in the captions.

Author's response: Fig. 4 has been deleted (XRF data and also #G. menardii are shown now in Fig. 3, plotted against age). We also note now that the Sr/Ca record has been truncated (l. 1666).

Reviewer's comment: Fig. 7E-F: Is it necessary to show G. ruber and G. sacculifer abundances from ODP Site 1063 here?

Author's response: Abundances of G. ruber and G. sacculifer from ODP Site 1063 have been removed.

Reviewer's comment: Reference: Bahr, A., Jiménez-Espejo, F.J., Kolasinac, N., Grunert, P., Hernández- Molina, F.J., Röhl, U., Voelker, A.H., Escutia, C., Stow, D.A. and Hodell, D. (2014) Deciphering bottom current velocity and paleoclimate signals from contourite deposits in the Gulf of Cádiz during the last 140 kyr: An inorganic geochemical approach. Geochemistry, Geophysics, Geosystems, 15, 3145-3160.

Author's response: References:

Bahr, A., Nürnberg, D., Karas, C. and Grützner, J.: Millennial-scale versus longterm dynamics in the surface and subsurface of the western North Atlantic Subtropical Gyre during Marine Isotope Stage 5, Glob. Planet. Change, 111, 77-87, doi:10.1016/j.gloplacha.2013.08.013, 2013.

Carew, J. L. and Mylroie, J. E.: Quaternary tectonic stability of the Bahamian archipelago: evidence from fossil coral reefs and flank margin caves, Quat. Sci. Rev., 14, 145-153, doi:10.1016/0277-3791(94)00108-N, 1995.

Chabaud, L., Ducassou, E., Tournadour, E., Mulder, T., Reijmer, J. J. G., Conesa, G., Giraudeau, J., Hanquiez, V., Borgomano, J. and Ross, L.: Sedimentary processes determining the modern carbonate periplatform drift of Little Bahama Bank, Mar. Geol., 378, 213-229, doi:10.1016/j.margeo.2015.11.006, 2016.

Please also note the supplement to this comment:
https://www.clim-past-discuss.net/cp-2018-38/cp-2018-38-AC1-supplement.pdf

[Figure]

Fig. 1. XRF-scan data from core MD99-2202.

**Fig. 1.**

[Figure]

Supplementary Figure 2 for the Author´s response

to the comments of A. Bahr

Fig. 2. Comparison of climatic records from core MD99-2202 and ODP Site 1058 (Bahr et al., 2013).

**Fig. 2.**

---

## Author Comment (AC2)

**Last interglacial ocean changes in the Bahamas: climate teleconnections between low and high latitudes**

Anastasia Zhuravleva[1] and Henning A. Bauch[2]

[1]Academy of Sciences, Humanities and Literature, Mainz, c/o GEOMAR Helmholtz Centre for Ocean Research, Wischhofstrasse 1-3, Kiel, 24148, Germany

[2]Alfred Wegener Institute, Helmholtz Centre for Polar and Marine Research c/o GEOMAR Helmholtz Centre for Ocean Research, Wischhofstrasse 1-3, Kiel, 24148, Germany

*Correspondence to*: Anastasia Zhuravleva (azhuravleva@geomar.de)

**Abstract.** Paleorecords and modeling studies suggest that instabilities in the Atlantic Meridional Overturning Circulation (AMOC) strongly affect the low-latitude climate, namely via feedbacks on the Atlantic Intertropical Convergence Zone (ITCZ). Despite pronounced millennial-scale climatic variability documented in the subpolar North Atlantic during the last interglacial (MIS 5e), studies on the cross-latitudinal teleconnections remain to be very limited, precluding full understanding of the mechanisms controlling subtropical climate evolution across the last warm cycle. Here, we present new planktic foraminiferal assemblage data combined with $\delta^{18}O$ values in surface and thermocline-dwelling foraminifera from the Bahama region, which is ideally suited to study past changes in subtropical ocean and atmosphere. Our data reveal that the peak sea surface warmth during early MIS 5e was intersected by an abrupt millennial-scale cooling/salinification event, which was possibly associated with a sudden southward displacement of the mean annual ITCZ position. This atmospheric shift, which could have left its imprint on the low-latitude upper ocean properties, is ascribed to the transitional climatic regime of early MIS 5e characterized by persistent ocean freshening in the high latitudes and, therefore, an unstable AMOC mode.

**1 Introduction**

In the low-latitude North Atlantic, wind patterns, precipitation-evaporation balance as well as sea surface temperatures (SSTs) and salinities (SSSs) are strongly dependent on the position of the Atlantic Intertropical Convergence Zone (ITCZ) and its associated rainfall (Peterson and Haug, 2006). Based on paleorecords and modelling studies, past positions of the ITCZ are thought to be related to the interhemispheric thermal contrast, changes of which could be driven by two principal mechanisms: the precessional cycle and, associated with it, a cross-latitudinal distribution of solar insolation and 
[revised manuscript text omitted]

Figure 3…: The age model for MIS 5Chronology…of…n core MD99-2202.… The temporal framework Age model…is based on alignment of (**b**) planktic δ¹⁸O values (Lantzsch et al., 2007) and (**D** …**d**) relative abundance record of *Globigerinoides* species and (**B**) planktic δ¹⁸O values (Lantzsch et al., 2007) …ith (**A…**) global benthic isotope stack LS16 (Lisiecki and Stern, 2016). (**C…**) Aragonite content in black (Lantzsch et al., 2007) and normalized elemental intensities of Sr in lilac and …s well as (**E…**) relative abundances of *G. menardii* and *G. menardii flexuosa* … [34]

Deleted: 2…: XRF-scan results, and…sedimentological and foraminiferal data from core MD99-2202 for the period 140-100 ka. (**a**) δ¹⁸O values in *G. ruber* (white); (**b**) aragonite content; (**a-C…**) is from Lantzsch et al. (2007). Normalized elemental intensities of (**D…**) Sr, (**e**) Ca and (**G…**) Cl, and…(**F…**) Sr/Ca intensity ratio (truncated at 0.6) and (**g**) absolute abundances of *G. menardii* per sample. Green bars denote core intervals with biased elemental intensities due to high [35]

Deleted: Figure 2: XRF-scan results and sedimentological data from core MD99-2202. (A) δ¹⁸O values in *G. ruber* (white); (B) aragonite content; (C) fraction with grain size <63 μm; (A-C) is from Lantzsch et al. (2007). Normalized elemental intensities of (D) Sr, (E) Ca and (G [36]

Deleted: 5…: Proxy records from core MD99-2202 over the last interglacial cycle. (**A…**) δ¹⁸O values in *G. ruber* (white) (Lantzsch et al., 2007), (**B…**) δ¹⁸O values in *G. truncatulinoides* (dex) (black) and *G. inflata* (blue), (**C… [37]

isotopic gradients between $\delta^{18}O$ values in *G. ruber* (white) and *G. truncatulinoides* (dex) and *G. ruber* (white) and *G. inflata*, respectively, (e-f) relative abundances of *G. inflata* and *G. truncatulinoides* (dex) respectively, (g) normalized Fe intensities. Also shown in (e) and (f) are modern relative foraminiferal abundances (average value ±1σ) around Bahama Bank, computed using 7 nearest samples from Siccha and Kučera (2017) database. T2 – Termination 2.

**Figure 5: Relative abundances of main *Globigerinoides* species in core MD99-2202 over the last interglacial cycle.** (a) $\delta^{18}O$ values in *G. ruber* (white) (Lantzsch et al., 2007), relative abundances of (b) *G. sacculifer*, (c) *G. ruber* (pink), (d) *G. conglobatus* and (e) *G. ruber* (white). Also shown in (b-e) are modern relative foraminiferal abundances (average value ±1σ) around Bahama Bank, computed using 7 nearest samples from Siccha and Kučera (2017) database. T2 – Termination 2.

**Figure 6: Comparison of proxy records from tropical, subtropical and subpolar North Atlantic over the last interglacial cycle.** (b) $\delta^{18}O$ values in *G. ruber* (white) in core MD99-2202 (Lantzsch et al., 2007), (c) relative abundances of the tropical species *G. sacculifer* and *G. ruber* (pink) in core MD99-2202, (d) molybdenum record from ODP Site 1002 (Gibson and Peterson, 2014), (e) $\delta^{13}C$ values measured in benthic foraminifera from core MD03-2664 (Galaasen et al., 2014, age model is from Zhuravleva et al., 2017b), (f) Ice-rafted debris in core PS1243 (Bauch et al., 2012, age model is from Zhuravleva et al., 2017b). Also shown is (a) boreal summer insolation (21 June, 30° N), computed with AnalySeries 2.0.8 (Paillard et al., 1996) using Laskar et al. (2004) data. Shown in (c) are modern relative abundances of *G. sacculifer* and *G. ruber* (pink) (average value ±1σ) around Bahama Bank, computed using 7 nearest samples from Siccha and Kučera (2017) database. The blue band suggests correlation of events (Younger Dryas-like cooling) across tropical, subtropical and subpolar North Atlantic (see text). T2 – Termination 2.

---

## Author Response (AR1)

- 4 Anastasia Zhuravleva and Henning A. Bauch

**5 A. Bahr (Referee)**

6 andre.bahr@geow.uni-heidelberg.de Received and published: 30 April 2018

Reviewer's comment: GENERAL REMARKS: The authors present a 7 comprehensive collection of faunal, stable isotope and sediment-geochemical 8 data from Little Bahama Bank (LBB) core MD99-2202 encompassing MIS 5e in 9 high temporal resolution. Such high-resolution low-latitude (27°N) records of the 10 penultimate Interglacial are rare, but important to constrain the climatic 11 variability of previous interglacials when compared to the Holocene. The authors 12 argue that the surface ocean variability at LBB reflects changes in the position of 13 the subtropical gyre and tropical warm pool, responding to latitudinal shifts of the 14 ITCZ that are driven by insolation and AMOC changes. In addition, the sea level 15 history at LBB is discussed, mainly based on the sedimentary composition 16 (aragonite content) of the sediment. In general, the author's interpretations are 17 well-founded by proxy evidence and supported by previous studies. Some 18 problematic aspects of the interpretation are discussed below, but do not interfere 19 with the general messages of the paper. 20

The manuscript is generally well-written and the study undoubtedly has its merits 21 as a valuable contribution for the understanding of low-latitude climate variability 22 during MIS 5e as well as the low-high-latitude feedbacks. However, the 23 manuscript lacks a clear focus. This regards in particular the introduction - it 24 should include more concise statements regarding the aims of the study, e.g. 25 hypotheses to be tested and specific questions that should be solved. At the 26 moment the introductory paragraphs (as well as the abstract ad conclusions) are 27 very general, partly with a focus that hinges strongly on local aspects of 28 sedimentary dynamics at LBB. Hence, I would strongly advocate to sharpen the 29 focus of the manuscript, as the reader is left with the impression that the study 30 confirms previous conceptual models (e.g. regarding the displacement of the 31 ITCZ during MIS 5e) but wonders about the specific take-home-messages and 32 new insights retrieved from this study. I therefore recommend the authors to re-33

- write the respective parts of their manuscript (in particular the introduction; see 34
- also specific comments below) to avoid underselling of their data. 35
- Author's response: 36

Please note that all changes in the manuscript are provided in a marked-up version 37

(track changes in Word, converted into a \*.pdf file). The line numbers used in the 38

39 current Author's response refer to the marked-up version, which is attached below.

- 40
- 41

The introduction has been rewritten, in attempt to provide a clearer focus of the 42

- study. Aims, methods and problematics are now also discussed. 43
- 44 **Reviewer's comment: SPECIFIC COMMENTS:**

Abstract: As discussed above, the abstract should be more specific about what 45 exactly the authors want to study. At present, the first three sentences concentrate 46 on the local/regional aspects concerning LBB, but in fact the data can be used to 47 infer much more general insights into low-high-latitude feedbacks and 48 subtropical gyre dynamics and Gulf Stream variability. Hence, I suggest to reduce 49 the reference to LBB but focus on the broader context. 50

Author's response: The rewritten abstract focuses on millennial-scale 51 teleconnections between high and low latitudes during the last interglacial. In 52 53 addition, the strong freshening in the high latitudes during early MIS 5e is described as the main reason for the warm-cold switches, observed across various 54 oceanic basins. 55

Reviewer's comment: Introduction: line 40 and elsewhere: I would avoid 56 abbreviating North Atlantic as "N. Atlantic" 57

Author's response: Done. 58

Reviewer's comment: 1. 50: ". . . . we attempt to close this gap. . ." reflects the 59 problem of the introduction - this is far too general. Data generation per se is 60 important, but should be done with some hypothesis/question to be tackled in 61 mind. At the moment I also miss a more specific lay out of the controversies that 62 are mentioned. This would help to formulate specific questions and hypotheses at 63 the end of the introduction. 64

- 65 Author's response: The introduction has been rewritten (see the comment above).
- 66 Reviewer's comment: Regional setting: 1. 64: capitalize "intertropical 67 convergence zone"
- 68 Author's response: Done (ls. 15-16, 69-70, 1651).
- Reviewer's comment: 1. 71: replace "tropical pool waters" with "tropical warmpool"
- Author's response: Replaced with "the Atlantic Pool Water" (ls. 117, 234).
- 72 Reviewer's comment: 1. 73: "thermocline layer" is too unspecific. Does this
- refer to the permanent thermocline?
- Author's response: Changed to the permanent thermocline (1. 237).
- Reviewer's comment: Methods: There should be a short statement in the introduction about the type of proxies used. In the present state, the purpose of the different proxies is unclear until the discussion. However, I would expect to read one or two sentences about the rational for XRF scanning, why  $\delta$ 18O of deep and shallow dwellers were used, and about the purpose of the faunal studies. Again, the mentioning of the proxies can be done in conjunction with the layout of the specific goals in the introduction (see comments above).
- Author's response: The used proxies are mentioned in the rewritten introduction
  together with their specific goals (ls. 119-128).
- Reviewer's comment: Results: 1. 163: "physical sediment properties" should be replaced by "sedimentological properties", this seems more appropriate as it refers to the grain size curve shown.
- Author's response: The text has been rephrased. Now "sedimentological records"
  are used (1. 389).
- 89 Reviewer's comment: 1. 174: I agree that the "significant sedimentological shift"
- 90 mentioned here is no artifact as it displays in different, independent proxies.
  91 However, given its minute amplitude relative to the general fluctuations in the
- core it is an overstatement to call it "significant". Considering the rather diffuse
- 93 discussion in Section 6.4 I would skip the reference and discussion of this feature
- 94 (see also respective comment below).

- Author's response: This section has been deleted from the results as well as from 95 the discussion. 96
- Reviewer's comment: 1. 178: "during the major deglacial transition . . . low 97 98 isotopic gradients. . ." this statement does not fully reflect the data, as there is a
- steady trend to more stratification from 135-129 ka, reaching the MIS 5e level of 99 well-stratified waters. As written in the text it sounds like the entire transition is
- 100
  - characterized by a persistent low isotopic gradients. 101
  - Author's response: We fully agree with the Reviewer's comment. The statement 102
- has been changed to "During the penultimate glacial maximum <...> gradients 103
- <...> are very low, succeeded by a gradually increasing difference across the T2, 104

~135-129 ka" (ls. 425-427). 105

- Reviewer's comment: 1. 181: please call out Fig. 6 after "species are observed" 106
- Author's response: Done (1.431). 107
- Reviewer's comment: 1. 185: please call out Fig. 5 after "abruptly increase". Also 108
- note that the variations of G. trunca (sin) in Fig. 5e are within the 1-sigma error 109
- of their present-day abundances. Is it necessary to plot these G. trunca (sin) 110
- abundances? 111
- Author's response: Fig. 5 (now Fig. 4) is called out after "together with a 112 reappearance of G. inflata" (ls. 433). For simplification and clarity, relative 113
- abundances of G. truncatulinoides (sin) as well as G. falconensis are not shown 114
- in figures any more. 115
- Reviewer's comment: Discussion: 1. 192-211: I wonder about the necessity to 116
- discuss the Sr/Ca record. In principle this is a good proxy for aragonite, however, 117
- the authors make the convincing case that this record is biased by changes in 118
- porosity and water content. Considering that the authors present the XRD-based 119
- record of aragonite form Lantzsch et al. (2007), the discussion of Sr/Ca can be 120
- omitted without losing information. 121
- Author's response: The discussion of our Sr/Ca record, in particular, of the 122
- "problematic intervals" is retained, however, has been substantially shortened (ls. 123
- 440-544). 124

- Reviewer's comment: 1. 222: Please add a reference for the subsidence rate of theLBB
- 127 Author's response: Done (study by Carew and Mylroie (1995) is cited, 1. 441)

128 Reviewer's comment: 1. 228: if I am correct, the sea level rise should be between

- 129 12-15 m (15 = 9 + 6 m) not 12-16 m. Please check.
- 130 Author's response: Correct. This paragraph has been, however, deleted.
- Reviewer's comment: 1. 256: "warm/cold conditions" please specify what is
  meant here.
- Author's response: Has been changed to "temperature estimations during late
  MIS 6 so far reveal cold subsurface conditions" (1. 670).
- 135 Reviewer's comment: 1. 270-272: In principle I agree with the interpretation that

136 G. truncatulinoides abundances strongly depend on the upper ocean stratification.

137 However, in this respect, it is interesting that G. trunca. (dex) is still high during

- 138 late MIS 5e, when  $\delta$ 18O is already low. Hence, vertical water column
- 139 stratification is not the sole factor influencing the G. trunc. abundances.
- 140 Author's response: We agree with the Reviewer's comment. A paragraph, dealing
- 141 with additional forcing factors controlling occurrences of G. truncatulinoides
- 142 (dex) and *G. inflata* has been included (ls. 688-803).
- 143 Reviewer's comment: 1. 283-287: Fe appears to lag  $\delta$ 18O, hence, question is if 144 dust is really the dominant factor that governs the Fe abundances if  $\delta$ 18O is 145 supposed to be the prime proxy recording for wind-driven water column 146 homogenization. Fe might also be influenced by diagenetic processes, hence, it
- 147 would be worthwhile looking at Ti/Al as Ti is not influenced by diagenesis.
- Author's response: We have reconsidered our Fe data and has significantly restricted the interpretation, given the "variety of additional effects that may have influenced our Fe-record" (ls. 816-820). Ti content in the investigated sediment
- 151 core appears to be very low, in addition, possibly strongly affected by seawater
- 152 content (Fig. 1).
- 153 Reviewer's comment: 1. 287-291: to check if winnowing plays a role during the
- 154 deglaciation elemental ratios such as Zr/Rb or Zr/Al might be used to check for
- 155 high bottom current velocities (Bahr et al., 2014).

- 156 Author's response: Increased winnowing at the northern slope of Little Bahama
- 157 Bank during glacial times (a result of an intensified wind-driven Antilles Current)
- 158 was previously suggested by Chabaud et al. (2016). As we could not prove the
- 159 statement by using Zr/Rb or Zr/Al data (Fig. 1), as suggested above, this part has
- 160 been removed.
- 161 Reviewer's comment: 1. 332: please add a reference after "only by ~124 ka"
- 162 Author's response: Done (l. 1081-1082).
- 163 Reviewer's comment: 1. 355: correct for "Hofman et al." (not Hofmann)
- 164 Author's response: Done. Changed for Hoffman et al.

165 Reviewer's comment: 1. 364: Notably, the *ruber* (w) abundances are strikingly

166 similar to the  $\delta$ 180ivf-sw record of *G*. *ruber* (w) from ODP Site 1058 (Bahr et

- 167 al, 2013). This supports the view that salinity is the main driver of G. *ruber* (w) abundances.
- 169 Author's response: We agree with the comment above, however, a common
- 170 temporal framework is needed for better core-to-core correlation and further
- 171 climatic implications (Fig. 2).
- 172 Reviewer's comment: 1. 372-381: as mentioned above, the discussed changes in 173 Sr and aragonite content are really minute compared to the other variations 174 observed in the proxy records. Given that the authors make only very general 175 inferences about the paleoclimatic implications I suggest to remove this 176 paragraph.
- 177 Author's response: Done. The paragraph has been removed.
- 178 Reviewer's comment: Conclusions 1. 387: "in the investigated core section": this
- 179 is much too local, especially for the conclusions (see also my general comments).
- 180 The broader implications of this study should become clear here.
- 181 Author's response: Conclusions have been rewritten to emphasize the broader
- 182 implications of the study (particularly, the third paragraph).
- 183 Reviewer's comment: 1. 389-392: these statements regarding the local
- 184 sedimentological processes on LBB are quite general and not novel considering
- 185 the amount of publications dealing with this topic.

- 186 Author's response: The statements have been removed from the conclusions.
- 187 Reviewer's comment: 1. 398: replace "depressed" by "shifted"
- 188 Author's response: Done (1. 1191).
- 189 Reviewer's comment: Figures: Fig. 4A-C is repetitive of Fig. 3 Fig. 4E: if Sr/Ca
- remains in the figure (see comments above): this record has been truncated at 0.3,
- 191 please state this in the captions.
- 192 Author's response: Fig. 4 has been deleted (XRF data and also #G. menardii are
- shown now in Fig. 3, plotted against age). We also note now that the Sr/Ca record
- 194 has been truncated (1. 1666).
- 195 Reviewer's comment: Fig. 7E-F: Is it necessary to show G. *ruber* and G. 196 *sacculifer* abundances from ODP Site 1063 here?
- Author's response: Abundances of *G. ruber* and *G. sacculifer* from ODP Site
  1063 have been removed.
- 199 Reviewer's comment: Reference: Bahr, A., Jiménez-Espejo, F.J., Kolasinac, N.,
- 200 Grunert, P., Hernández- Molina, F.J., Röhl, U., Voelker, A.H., Escutia, C., Stow,
- 201 D.A. and Hodell, D. (2014) Deciphering bottom current velocity and paleoclimate
- signals from contourite deposits in the Gulf of Cádiz during the last 140 kyr: An
- 203 inorganic geochemical approach. Geochemistry, Geophysics, Geosystems, 15,
- 204 3145-3160.
- 205 Author's response: References:

Bahr, A., Nürnberg, D., Karas, C. and Grützner, J.: Millennial-scale versus longterm dynamics in the surface and subsurface of the western North Atlantic
Subtropical Gyre during Marine Isotope Stage 5, Glob. Planet. Change, 111, 77–
87, doi:10.1016/j.gloplacha.2013.08.013, 2013.

- 210 Carew, J. L. and Mylroie, J. E.: Quaternary tectonic stability of the Bahamian
- archipelago: evidence from fossil coral reefs and flank margin caves, Quat. Sci.
  Rev., 14, 145–153, doi:10.1016/0277-3791(94)00108-N, 1995.
- 213 Chabaud, L., Ducassou, E., Tournadour, E., Mulder, T., Reijmer, J. J. G., Conesa,
- 214 G., Giraudeau, J., Hanquiez, V., Borgomano, J. and Ross, L.: Sedimentary

- 215 processes determining the modern carbonate periplatform drift of Little Bahama
- 216 Bank, Mar. Geol., 378, 213–229, doi:10.1016/j.margeo.2015.11.006, 2016.

217

**218 Author's response to the comments of the Anonymous Referee #2**

- 219 Interactive comment on "The last interglacial (MIS 5e) cycle at Little
- 220 Bahama Bank: A history of climate and sea-level changes" by
- 221 Anastasia Zhuravleva and Henning A. Bauch
- 222 Anonymous Referee #2
- 223 Received and published: 17 May 2018
- 224 Reviewer's comment: SUMMARY: Zhuravleva and Bauch present a detailed
- 225 consideration of the climate evolution of the Last Interglacial (LIg) for a core site
- on the Little Bahama Bank (LBB) using faunal assemblage and scanning XRF
- 227 techniques. The high resolution faunal assemblages nicely resolve hydrographic
- oscillations at the site for the LIg reflecting both the insolation driven and AMOC
- 229 modulated migration of the ITCZ for this region.
- 230 I would recommend the following amendments/clarifications: (a) change of title
- 231 to better reflect the content of the paper; (b) removal or at the very least
- restructuring of the discussion of sea level. This section could be significantly
- trimmed and simplified (no new insights offered but a nice corroboration).
- Alternatively, if the authors wish to retain the sea-level discussion, then discussion of other sea level evidence from the region, glacio-isostatic adjustment
- 236 (GIA) processes etc. are needed. (c) clearer discussion of the teleconnections
- between N Atlantic oceanic changes (i.e., variation in AMOC), the migration of
- the ITCZ and surface hydrographic change at MD99-2202.
- 239 Author's response:
- Please note that all changes in the manuscript are provided in a marked-up version
  (track changes in Word, converted into a \*.pdf file). The line numbers used in the
  current Author's response refer to the marked-up version, which is attached
  below.
- 244
- a) The title has been changed to better reflect the main finding of the paper,
  i.e., subpolar forcing on the subtropical climate during the last interglacial;
- b) The discussion about sea level is significantly reduced and focuses now exclusively on relative sea level changes in the Bahama region and its implications for regional sedimentary processes (ls. 440-544);

- c) Links between AMOC strength and ITCZ shifts are discussed now in the
  introduction (ls. 68-78) as well as briefly mentioned in the discussion (ls.
  986-1078).
- Reviewer's comment: GENERAL COMMENTS: The manuscript, in general,
  reads well. However, the structure and focus of the paper requires further thought.

255 A clear statement of the research questions was missing and is reflected in the

- 256 general tone of the introduction (and the manuscript generally).
- Author's response: The introduction has been rewritten, in attempt to outline the aims of the manuscript (ls. 95-110), used proxies (ls. 119-128) and testing hypothesis.
- 260 Reviewer's comment: 1. Title

I found this to be somewhat misleading. The data in Zhuravleva and Bauch is not 261 a sea-level record per se, rather a record of increased aragonite supply to the core 262 site during interglacials, with these intervals of increased aragonite 263 production/supply likely corresponding to < -6 m relative sea level (RSL) due to 264 the generally shallow nature of Little Bahama Bank (i.e., you can infer periods of 265 <-6 m relative sea level). This work nicely corroborates the Lantzsch et al., 2007 266 and Chabaud et al., 2016 studies but isn't a sea-level story. What is new and 267 interesting the palaeoceanographic evolution of the Last Interglacial (LIg) at the 268 site, and the interplay of interglacial climate (movement of the ITCZ etc.). I would 269 suggest changing the title to better reflect this. 270

- 271 Author's response: The title has been changed.
- 272 Reviewer's comment: 2. Sea level

273 This section requires some restructuring to help the reader. The definition of the "flooding interval" (and corresponding relative sea level, <-6 m) is key to this 274 section of the manuscript but I struggled to clearly follow the logic of how you 275 defined the flooding interval using your records and why a -6 m RSL for this 276 interval was appropriate. The connection between the flooding interval and 277 inferred RSL of < -6 m was found al- most at the end of the section (line 222 to 278 226) when it should be at the start. All the information is there but the reader has 279 to work hard to follow the argument. 280

281 Perhaps something along the lines of;

modern LBB lagoon is shallow, with an average water depth < 6 m (Williams,</li>
1985); tectonic stability of the region (refs needed);

during the LIg, increasing RSL at the site floods the generally shallow bank and
increases the area for aragonite production (i.e., the carbonate shedding model,
Droxler and Schlager, 1985; Schlager et al., 1994);

- 287 Conversely, during glacial intervals, the top is exposed which limits the 288 production and export of aragonite;
- As such, we define the flooding interval (and inferred <-6 m RSL) is defined by
- 290 an increase in the sedimentation rate, increase in wt % aragonite, increased Sr/Ca
- ratio, increase % *Globigerinoides*/decrease in numbers of *G. menardii*.

This could then usefully be followed with your discussion of very high values of Sr/Ca due to increased saltwater (lines 192 to 211). Perhaps shade these 'problematic' Sr/Ca intervals in subsequent figures? You should also note the truncation of the Sr/Ca record in caption of Figure 4.

296 I would suggest confining discussion of sea level in this section to that suggested above. If you wish to make more of the sea level story, then greater consideration 297 of other Bahamas sea-level records, as well as those from the wider area is 298 needed. For example, the +6m notch on Little Sale Cay (LLB) (Neumann and 299 Hearty, 1996), other geomorphological records (e.g., Hearty and Kindler, 1995; 300 Neumann and Hearty, 1996), the elevated Last Interglacial (LIg) coral records of 301 Chen et al 1991, Hearty et al., 2007, Thompson et al., 2011 and the regionally 302 extensive erosional surface that is suggestive of a sea-level oscillation within the 303 LIg (Bahamas, Florida and Yucatan; Chen et al 1991, Hearty et al., 2007, 304 Blanchon et al., 2009; Thompson et al., 2011). 305

How does the timing of the highstand from the coral/other records from the Bahamas compare to the timing of the interval of enhanced aragonite production (and inferred sea levels < -6 m)? How do changes in hydrography (variations in faunal assemblages) at the site compare to the timing of the Bahamas LIg highstand? The broad correspondence between climate ( $\delta$ 18OG.ruber) and relative sea level (RSL) (weight % aragonite) is hinted at in lines 138 to 141 but could be developed further if you wish to keep the sea-level discussion.

Any discussion of the LIg highstand in a general sense (i.e., the eustatic record) (lines 227 to 231) and Bahamas RSL will need to consider glacio-isostatic (GIA)

- 315 processes, given the intermediate location of the site on the peripheral bulge of
- the former Lauren- tide Ice Sheets. There will be a regional expression of the LIg
- 317 highstand; the Bahamas would "see" a "late" LIg highstand compared to eustatic

318 sea level (e.g., Figure 6 in Stirling et al., 1998). There seems to be good

319 correspondence between the age of your "flooding interval" at the site (i.e., sea

- level < -6 m) and the predictions of RSL (Stirling et al., 1998, their Figure 6).
- 321 Given that the records presented are not strictly a sea level record, rather
- 322 incidence of increased aragonite production/export, and seems to corroborate
- 323 previous studies rather than adding anything new, I would confine this section to
- 324 just a brief consideration of the timing of your "flooding interval".
- 325 Author's response: We agree on the importance of consideration of glacio-326 isostatic adjustment processes for interpretation of our aragonite-related records
- 327 in terms of eustatic sea level change and also for comparison with other sea level
- 328 reconstructions and curves (ls. 746-749). Therefore, we have restructured the sea-
- 329 level discussion, in accordance with the Reviewer's comment and significantly
- 330 reduced this part, restricting the discussion to the relative sea level change,
- 331 defining the "flooding interval" and associated changes in geochemical and
- 332 sedimentary data around Bahama Banks (ls. 440-544).
- The study by Carew and Mylroie (1995) was cited with regard to the tectonic stability of the Bahama region (1. 441). Truncation of the Sr/Ca record is now mentioned in the figure caption (Fig. 3, 1. 1666).
- 336 Reviewer's comment: 3. Palaeoceanographic reconstruction
- This section is much more coherent and well written. I would recommend this as the focus of the manuscript.

The discussion, while nicely documenting the site/regional changes during the 339 LIG, was lacking in consideration of the mechanisms. This section would be 340 strengthened by a clearer exposition of the mechanisms linking ITCZ position, 341 insolation (precession and the migration of the ITCZ to the warming hemisphere) 342 and AMOC (i.e., modification of the thermal condition at the surface, due to 343 interactions with the ocean, that in turn act to drive atmospheric circulation). This 344 is well documented for the last deglaciation and glacial period (and in modelling 345 studies), where N. hemisphere extra-tropical cooling (brought about by variations 346 in the AMOC, forced by freshwater inputs) lead to an interhemispheric thermal 347

348 gradient and a southward shift in the ITCZ (e.g., review of Chiang and Friedman,

349 2012 or Schneider et al., 2014). This would help the reader to place the different

350 records (Cariaco, MD99-2202 and Site 1063) within a broader climatological

- 351 context. Again, all the 'threads' of the story are there, it just needs a stranger
- 352 framework.

For example, I found the correspondence between the % *G. ruber* and *G. sacculifer* and the XRF Mo count of the Caricao Basin striking (demonstrating the clear record of ITCZ shifts at the LBB) but the link to N. Atlantic surface density changes (AMOC slowdown with surface freshening e.g., Galaasen et al., 2014 etc.) and positive the  $\delta 180 G$ . *ruber* excursion and faunal changes at MD99-2202 and southward migration of the ITCZ (Cariaco Mo, MD99-2202 decrease

- in % *Globigerinoides*) weak. A short introductory paragraph should fix this.
- 360 Author's response: We have included information on coupling between high-

361 latitude forcing (AMOC strength) on the ITCZ position with its further influence

362 on upper ocean properties in the introduction (ls. 68-78) and as well as briefly in

363 the discussion (ls. 986-1078).

Reviewer's comment: A southward shift of the ITCZ, strengthens of the trade winds increases the eolian input from the Sahara, resulting in reduced Al/Ti in Cariaco. These episodes of decreased Al/Ti ratios in Cariaco correspond to elevated salinities in the Caribbean (e.g., Yarincik et al 2000). I assume there is clear correspondence between the Cariaco Al/Ti and Mo records and hence to your % of tropical species?

- Author's response: Despite similar approach for age model construction (alignment of stable isotope data to SPECMAP/benthic stack), there is no correspondence between the Al/Ti record from Yarincik et al. (2000) and the Modata from Gibson and Peterson (2014) and, therefore, our relative abundance of the tropical species. This is likely due to low-resolution of the first record, providing general information about atmospheric circulation changes mainly on
- 376 glacial-interglacial timescales (Fig. 1).
- 377 Reviewer's comment: Do you see an increase in iron (with increased dust
- transport) in your record during the positive the  $\delta$ 180G.ruber excursion ~ 127
- ka? (plotting this on a log scale for the LIg might help).

Author's response: We don't find any response in iron accumulation during the127-ka event.

382 Reviewer's comment: Dust inputs are probably better reflected in the XRF core

scanning Ti record, given that your Fe inputs could change with a number of factors

384 factors.

Author's response: We agree with this statement, but our XRF Ti measures appear to be very low and could be strongly influenced by Cl content (Fig. 2), therefore they were not considered in the study. We agree on the comment and restrict the

- interpretation of our Fe content, due to the "variety of additional effects that may
- have influenced our Fe-record" (ls. 816-820).
- 390 Reviewer's comment: It would be interesting to compare to your faunal 391 assemblages and a calculated  $\delta$ 18Oseawater for MD99-2202.
- 392 Author's response: Please, see further below.
- Reviewer's comment: Additionally, is there any correspondence to the dated palaeosols on the Bahamas (Muhs et al., 2007)?

Author's response: Study of Muhs et al. (2007) reveals two major sources for the dated palaeosols (~125 ka) on the Bahamas: African dust and Mississippi River valley loess. Today particularly strong input of African aerosols occurs during summer time, when the ITCZ position is to the north. The study, thus, suggests variable parent materials for eolian inputs, possibly with a greater role of Mississippi River valley loess at times of southward ITCZ shifts (glaciations).

401 The text has been improved accordingly (ls. 816-820).

402 Reviewer's comment: Given you have faunal assemblage data, could you 403 calculate a transfer function/MAT sea surface temperature? From this you could 404 then calculate  $\delta$ 18Oseawater at the site to think about density changes during the 405 LIg.

Author's response: While we agree with the suggestion to look at density changes and inferred calculated salinities, we assume that the use of Mg/Ca-based temperatures, derived from similar species used to obtain d18O values, would be more plausible. As we don't have Mg/Ca-ratios for the investigated samples, we rather suggest considering proportions of selected species for relative temperature/salinity change reconstructions.

- 412
- 413 Reviewer's comment: lines 372 to 381 I found this paragraph to be highly
- 414 speculative and not well supported by your data (I struggled to see the change in
- the sedimentological properties in your figures, even with the help of the white
- 416 arrows). I would suggest removing this section.
- 417 Author's response: The section has been removed.
- 418 Reviewer's comment: TECHNICAL CORRECTIONS
- 419 Referencing: greater care needed with referencing. Please check manuscript. For
- 420 example, the depth of submersion of the LBB (and origin of the -6 m quoted
- 421 often in the paper) is Neumann and Land, 1975. Similarly the carbonate

422 shedding model (upon which the inferred sea-level story is based) comes from

- 423 Droxler and Schlager, 1985; Schlager et al., 1994
- 424 Author's response: We tried to improve the referencing. Neumann and Land,
- 425 1975 and Droxler and Schlager, 1985 are cited (l. 444 and ls. 244, 250, 283,
- 426 444, respectively).
- 427 Reviewer's comment: line 124 what potential biases are you refereeing to?
  428 Please give appropriate references for these.
- 429 Author's response: PhD thesis of Chabaud, 2016 has been cited (1. 336). This
- 430 study extensively discusses the potential biases for XRF measurements in
- 431 periplatform sediments, related e.g., to coarse-grained intervals, increased
- 432 sediment porosity and/or seawater content.
- 433 Reviewer's comment: line 242 what is the derivation of "minimal ice volume
- 434 interval" and it's reference?
- 435 Author's response: This part has been removed.
- 436 Reviewer's comment: line 332 reference required for the "full resumption of the
- 437 AMOC. . . only by ~124 ka"
- 438 Author's response: Studies of Hodell et al. (2009) and Barker et al. (2015) are 439 cited (1. 1081-1082).
- 440 Reviewer's comment: 2. Other: line 64 capitalisation of "intertropical
- 441 convergence zone"

- 442 Author's response: Done (ls. 15-16, 69-70, 1651).
- 443 Reviewer's comment: line 133 unit of measurement missing, add "m".
- 444 Author's response: Done (1. 344).

445 Reviewer's comment: line 330 - please clarify or add examples of the "additional446 forcing" or add "as discussed below".

Author's response: The term "additional forcing" (now changed to "other forcing
factors at play", 1. 986) is clarified (i.e., AMOC control on the ITCZ position, ls.
986-1078).

450 Reviewer's comment: line 324-325 - Clarification of the age of the 451 cooling/increase salinity event ~ 127 ka is needed. The ~127 ka age for this event 452 is derived from your age model, whereas the U-series ages for the correlative 453 event in core 152JPC (Bahamas, Slowey et al., 1996), dated (in duplicate) above 454 and below the event to ~121 ± 3 ka and 125.6 ka (mean of  $127 \pm 4.8$  and  $124.1 \pm$ 455 5.1) respectively. (Note, the relatively large age uncertainties associated with 456 these U-series ages)

457 Author's response: We agree with the Reviewer's comment both with regard to 458 only subtle agreement between our age estimates and direct U-Th dating of the 459 cooling/salinification event, as well as large age uncertainties associated with U-460 series ages. Therefore, this section has been removed.

461 Reviewer's comment: 3. Figures: typo - Axis label in Figure 4D should read "#

462 G. menardii" rather than "# G. menradii" I would remove the sea level records

(H), or if you choose to retain these and your discussion of sea level, then you
should remove the dashed blue line "RSL above -6 m" from the lowermost panel
(H).

- 466 Author's response: Fig. 4 has been removed, instead XRF data and # G. menardii467 are shown together in Fig. 3, plotted against age.
- 468
- 469 Reviewer's comment: REFERENCES CITED: Blanchon, P., Eisenhauer, A.,
- 470 Fietzke, J. and Liebetrau, V., 2009. Rapid sea-level rise and reef back-stepping at
- the close of the last interglacial highstand. Nature, 458, 881.

- 472 Chabaud, L., Ducassou, E., Tournadour, E., Mulder, T., Reijmer, J. J. G., Conesa,
- 473 G., Giraudeau, J., Hanquiez, V., Borgomano, J. and Ross, L.: Sedimentary
- 474 processes determining the modern carbonate periplatform drift of Little Bahama
- 475 Bank, Mar. Geol., 378, 213–229.
- 476 Chen, J.H., Curran, H.A., White, B., Wasserburg, G.J. 1991. Precise chronology
- 477 of the last interglacial period: 234U-230Th data from fossil coral reefs in the
- 478 Bahamas. Geological Society of America Bulletin 103, 82-97.
- Chiang, J. C. H. & Friedman, A. R. Extratropical cooling, interhemispheric
  thermal gradients, and tropical climate change. Annu. Rev. Earth Planet. Sci. 40,
  383–412 (2012)
- 482 Diz et al 2018. Ocean and atmosphere teleconnections modulate east tropical
- 483 Pacific productivity at late to middle Pleistocene terminations. Erath and
- 484 Planetary Science Letters 49, 82-91.
- 485 Droxler, A.W., Schlager, W., 1985. Glacial versus interglacial sedimentation
  486 rates and tur- bidite frequency in the Bahamas. Geology 13, 799–802.
- 487 Galaasen, E. V., Ninnemann, U. S., IrvalÄs´, N., Kleiven, H. (Kikki) F.,
  488 Rosenthal, Y., Kissel, C. and Hodell, D. A.: Rapid Reductions in North Atlantic
  489 Deep Water During the Peak of the Last Interglacial Period, Science, 343, 1129.
- Hearty PJ, Kindler P. Sea-level highstand chronology from stable carbonate
  platforms (Bermuda and the Bahamas). Journal of Coastal Research. 1995 Jul
- 492 1:675-89.
- 493 Hearty, P.J., Hollin, J.T., Neumann, A.C., O'Leary, M.J. and McCulloch, M.,
- 494 2007. Global sea-level fluctuations during the Last Interglaciation (MIS 5e).
- 495 Quaternary Science Reviews, 26, 2090-2112.
- 496 Lantzsch,H.,Roth,S.,Reijmer,J.J.G.andKinkel,H.:SeaâA R levelrelatedresedi-
- 497 mentation processes on the northern slope of Little Bahama Bank (Middle498 Pleistocene to Holocene), Sedimentology, 54, 1307–1322.
- 499 Mahowald, N. M., D. R. Muhs, S. Levis, P. J. Rasch, M. Yoshioka, C. S. Zender,
- 500 and C. Luo (2006), Change in atmospheric mineral aerosols in response to
- 501 climate: Last glacial period, preindustrial, modern, and doubled carbon dioxide
- 502 climates, J. Geophys. Res., 111, D10202, doi:10.1029/2005JD006653.

- Muhs, D.R., Budahn, J.R., Prospero, J.M., Carey, S.N. 2007. Geochemical
  evidence for African dust inputs to soils of western Atlantic islands: Barbados,
  the Bahamas, and Florida. Journal of Geophysical Research 112, F02009
  doi:10.1029/2005JF000445.
- Neumann, C.A. & Hearty PJ. 1996. Rapid sea-level changes at the close of the
  last interglacial (substage 5e) recorded in Bahamian island geology. Geology,
  24(9):775-8.
- 510 Neumann, A.C., Land, L., 1975. Lime mud deposition and calcareous algae in the
- 511 bight of Abaco, Bahamas: a budget. J. Sediment. Petrol. 45, 763–786.
- 512 Schlager, W., Reijmer, J.J.G., Droxler, A.W., 1994. Highstand shedding of 513 carbonate plat- forms. J. Sediment. Res. B64, 270–281.
- 514 Schneider, T., Bischoff, T. & Haug, G. H. Migrations and dynamics of the 515 intertropical convergence zone. Nature 513, 45–53 (2014).
- Slowey, N.C., Henderson, G.M., Curry, W.B. 1996. Direct U–Th dating of
  marine sediments from the two most recent interglacial periods. Nature 382, 242244.
- 519 Stirling, C.H., Esat, T.M., Lambeck, K., McCulloch, M.T. 1998. Timing and
- 520 duration of the Last Interglacial: evidence for a restricted interval of widespread
- 521 coral reef growth. Earth and Planetary Science Letters 160, 745–762.
- 522 Thompson, W.G., Curran, H.A., Wilson, M.A. and White, B., 2011. Sea-level
- 523 oscilla- tions during the last interglacial highstand recorded by Bahamas corals.
- 524 Nature Geo- science, 4, 684-687.
- 525 Williams, S.C., 1985. Stratigraphy, Facies Evolution and Diagenesis of Late
- 526 Cenozoic Lime- stones and Dolomites, Little Bahama Bank, Bahamas. Univ.
- 527 Miami, Coral Gables FL (217 pp.).
- 528 Yarincik, K.M., Murray, R.W. and Peterson, L.C., 2000. Climatically sensitive
- 529 eolian and hemipelagic deposition in the Cariaco Basin, Venezuela, over the past
- 530 578,000 years: Results from Al/Ti and K/Al. Paleoceanography, 15(2), 210-228.
- 531
- 532

- 533 Author's response: References cited:
- Barker, S., Chen, J., Gong, X., Jonkers, L., Knorr, G. and Thornalley, D.: Icebergs not the trigger for North Atlantic cold events. *Nature*, *520*(7547), 333, 2015.
- 536 Carew, J. L. and Mylroie, J. E.: Quaternary tectonic stability of the Bahamian 537 archipelago: evidence from fossil coral reefs and flank margin caves, *Quat. Sci.*
- 538 *Rev.*, 14, 145–153, doi:10.1016/0277-3791(94)00108-N, 1995.
- 539 Droxler, A.W. and Schlager, W.: Glacial versus interglacial sedimentation rates 540 and tur- bidite frequency in the Bahamas. *Geology* 13, 799–802, 1985.
- Gibson, K. A. and Peterson, L. C.: A 0.6 million year record of millennial-scale
  climate variability in the tropics, *Geophys. Res. Lett.*, 41, 969–975,
  doi:10.1002/2013GL058846, 2014.
- Hodell, D. A., Minth, E. K., Curtis, J. H., McCave, I. N., Hall, I. R., Channell, J.
- 545 E. and Xuan, C.: Surface and deep-water hydrography on Gardar Drift (Iceland
- Basin) during the last interglacial period. *Earth Planet. Sci. Let.*, 288(1-2), 10-19,
  2009.
- Muhs, D.R., Budahn, J.R., Prospero, J.M. and Carey, S.N.: Geochemical evidence
  for African dust inputs to soils of western Atlantic islands: Barbados, the
  Bahamas, and Florida. J. Geophys. Res. 112, F02009 doi:10.1029/2005JF000445,
- 551 2007.
- Neumann, A.C. and Land, L., 1975. Lime mud deposition and calcareous algae in the bight of Abaco, Bahamas: a budget. *J. Sediment. Petrol.* 45, 763–786.
- Yarincik, K. M., Murray, R. W. and Peterson, L. C.: Climatically sensitive eolian
  and hemipelagic deposition in the Cariaco Basin, Venezuela, over the past
  578,000 years: Results from Al/Ti and K/Al. *Paleoceanogr. Paleoclim.*, *15*(2),
  210-228, 2000.

| Last interglacial ocean changes in the Bahamas: climate                                                                  |      | Deleted                |
|--------------------------------------------------------------------------------------------------------------------------|------|------------------------|
| teleconnections between low and high latitudes                                                                           |      | Deleted                |
|                                                                                                                          |      | Deleted                |
| Υ                                                                                                                        |      | Deleted                |
| Anastasia Zhuravleva 1 and Henning A. Bauch 2                                                      |      | Deleted                |
| 1 Academy of Sciences, Humanities and Literature, Mainz, c/o GEOMAR Helmholtz Centre for Ocean Research,      |      |                        |
| Wischhofstrasse 1-3, Kiel, 24148, Germany                                                                                |      |                        |
| 2 Alfred Wegener Institute, Helmholtz Centre for Polar and Marine Research c/o GEOMAR Helmholtz Centre for    |      |                        |
| Ocean Research, Wischhofstrasse 1-3, Kiel, 24148, Germany                                                                |      |                        |
|                                                                                                                          |      |                        |
| Correspondence to: Anastasia Zhuravleva (azhuravleva@geomar.de)                                                          |      |                        |
|                                                                                                                          |      |                        |
|                                                                                                                          |      | Deleted                |
| Abstract. Paleorecords and modeling studies suggest that instabilities in the Atlantic Meridional Overturning            |      | investiga
level cha |
| Circulation (AMOC) strongly affect the low-latitude climate, namely via feedbacks on the Atlantic Intertropical          |      | Deleted                |
| Convergence Zone (ITCZ). Despite pronounced millennial-scale climatic variability documented in the subpolar             |      | Deleter                |
| North Atlantic during the last interglacial (MIS 5e), studies on the cross-latitudinal teleconnections remain to be      |      | northern
that the b |
| very limited, precluding full understanding of the mechanisms controlling subtropical climate evolution across           | $\ $ | "plateau"
-6 m for  |
| the last warm cycle. Here, we present new planktic foraminiferal assemblage data combined with $\delta^{18}$ O values in |      | which to
intertrop  |
| surface and thermocline-dwelling foraminifera from the Bahama region, which is ideally suited to study past              | $\ $ | During e               |
| changes in subtropical ocean and atmosphere. Our data reveal that the peak sea surface warmth during early MIS           |      | Atlantic
early MI   |
| 5e was intersected by an abrupt millennial-scale cooling/salinification event, which was possibly associated with        |      | millennia
Deleted   |
| a sudden southward displacement of the mean annual ITCZ position. This atmospheric shift, which could have               |      | Deleted                |
| left its imprint on the low-latitude upper ocean properties, is ascribed to the transitional climatic regime of early    |      | Deleted                |
| MIS 5e characterized, by persistent ocean freshening in the high latitudes and, therefore, an unstable AMOC              | Ľ    | Deleted                |
| mode                                                                                                                     |      | Deleted                |
| v                                                                                                                        |      | Delete
persuas      |
|                                                                                                                          |      | volume c               |

1

**С**

| (   | Deleted: The last interglacial (MIS 5e) |
|-----|-----------------------------------------|
| (   | Deleted: and itsimpact on               |
| ~(  | Deleted: cycle at Little Bahama Bank:   |
| X   | Deleted: the                            |
| . 1 |                                         |

A history of climate and sea-level changes

Shallow-water sediments of the Bahama region the last interglacial (MIS 5e) are ideal to the region's sensitivity to past climatic and sea ges.

**new faunal**

shifts in the ITCZ

isotopic and XRF-sediment core data from the , isotopic and XRF-sediment core data from the lope of the Little Bahama Bank. The results suggest nk top remained flooded across the last interglacial ~129-117 ka, arguing for a relative sea level above is time period. In addition, climatic variability, ay is closely coupled with movements of the al convergence zone (ITCZ), is interpreted based sotopes and foraminiferal assemblage records. 1y MIS 5e, the mean annual ITCZ position moved in line with increased solar forcing and a recovered leridional Overturning Circulation (AMOC). The 5e warmth peak was intersected. however, by a 5e warmth peak was intersected, however, by a -scale cooling ev

| 1          | Deleted: tropical      |
|------------|------------------------|
| Â          | Deleted: ,             |
| $/\lambda$ | Deleted: high-latitude |
| 2          | Deleted: thereby       |
| 1          | Deleted: ,             |
| -(         | Deleted:               |
|            | 7                      |

Our shallow-water records from the Bahamas ly demonstrate that not only was there a tight tween last interglacial sea level history and ice anges, via the atmospheric forcing we could further ra-interglacial connectivity between the polar and latitudes that left its imprint also on the ocean circulation

| 67 | 1 Introduction                                                                                                         |      | Deleted: ¶                                        |
|----|------------------------------------------------------------------------------------------------------------------------|------|---------------------------------------------------|
| 68 | In the low-latitude North Atlantic, wind patterns, precipitation-evaporation balance as well as sea surface            |      |                                                   |
| 69 | temperatures (SSTs) and salinities (SSSs) are strongly dependent on the position of the Atlantic Intertropical         |      |                                                   |
| 70 | Convergence Zone (ITCZ) and its associated rainfall (Peterson and Haug, 2006). Based on paleorecords and               |      |                                                   |
| 71 | modelling studies, past positions of the ITCZ are thought to be related to the interhemispheric thermal contrast,      |  | Deleted: is                                       |
| 72 | changes of which could be driven by two principal mechanisms: the precessional cycle and, associated with it, a        |      | Deleted: strongly                                 |
| 73 | cross-latitudinal distribution of solar insolation and millennial-scale climatic variability brought about by Atlantic |      |                                                   |
| 74 | Meridional Overturning Circulation (AMOC) instabilities (Wang et al., 2004; Broccoli et al., 2006; Arbuszewski         |      |                                                   |
| 75 | et al., 2013; Schneider et al., 2014). Specifically, millennial-scale cold events in the high northern latitudes were  |      | Deleted: are associated                           |
| 76 | linked with reduced convection rates of the AMOC, accounting for both a decreased oceanic transport of the             |      |                                                   |
| 77 | tropical heat towards the north and a southward shift of the mean annual position of the ITCZ (Vellinga and Wood,      |      | Deleted: ,                                        |
| 78 | 2002; Chiang et al., 2003; Broccoli et al., 2006). Reconstructions from the low-latitude North Atlantic confirm        |      | Deleted: are consistent with                      |
| 79 | southward displacements of the ITCZ coeval with AMOC reductions and reveal a complex hydrographic response             | ſ    | Deleted: s                                        |
| 80 | within the upper water column, generally suggesting an accumulation of heat and salt in the (sub)tropics (Schmidt      |      |                                                   |
| 81 | et al., 2006a; Carlson et al., 2008; Bahr et al., 2011; 2013). There are, however, opposing views on the subtropical   |      |                                                   |
| 82 | sea surface development at times of high-latitude cooling events, While some studies suggest stable or increasing      |      | Deleted: coolings                                 |
| 83 | SSTs (Schmidt et al., 2006a; Bahr et al., 2011; 2013), others imply an atmospheric-induced (evaporative) cooling       |      | Deleted: :                                        |
| 84 | (Chang et al., 2008; Chiang et al., 2008).                                                                             | J    | Deleted: w                                        |
| 85 | The last interglacial (MIS 5e), lasting from about ~130 to 115 thousand years before present (hereafter [ka]), is      |      |                                                   |
| 86 | often referred to as a warmer-than-preindustrial interval, associated with significantly reduced ice sheets and a      |      |                                                   |
| 87 | sea level rise up to 6-9 meters above the present levels (Dutton et al., 2015; Hoffman et al., 2017). This time        |      |                                                   |
| 88 | period has attracted a lot of attention as a possible analog for future climatic development as well as a critical     |      |                                                   |
| 89 | target for validation of climatic models (Masson-Delmotte et al., 2013). Proxy data from the North Atlantic            |      |                                                   |
| 90 | demonstrate that the climate of the last interglacial was relatively unstable, involving one or several cooling events |      |                                                   |
| 91 | (Maslin et al., 1998; Fronval and Jansen, 1997; Bauch et al., 2012; Irvali et al., 2012, 2016; Zhuravleva et al.,      |      |                                                   |
| 92 | 2017a, b). This climatic variability is thought to be strongly related to changes in the AMOC strength (Adkins et      |      |                                                   |
| 93 | al., 1997). Thus, recent studies reveal that the AMOC abruptly recovered after MIS 6 deglaciation (Termination         |      | Deleted: during MIS6 deglaciation (Termination 2) |
| 94 | 2 or T2), i.e., at the onset of MIS 5e, at ~ 129 ka, but it was interrupted around 127-126 ka (Galaasen et al., 2014;  |      |                                                   |
| 95 | Deaney et al., 2017). Despite the pronounced millennial-scale climatic variability documented in the high northern     |      |                                                   |
| 96 | latitudes, studies on the cross-latitudinal links are very limited (but see e.g., Cortijo et al., 1999; Schwab et al., |      |                                                   |

| 108 | 2013; Kandiano et al., 2014; Govin et al., 2015; Jiménez-Amat and Zahn, 2015). This precludes the full               |                                                                                                                                                                                                                                                                                                                                                                                                                                                                                                                                                                                                                                                                                                                                                                                                                                                                                                                                                                                                                                                                                                                                                                                                                                                                                                                                                                                                                                                                                                                                                                                                                                                                                                                                                                                                                                                                                                                                                                                                                                                                                                                                |
|-----|----------------------------------------------------------------------------------------------------------------------|--------------------------------------------------------------------------------------------------------------------------------------------------------------------------------------------------------------------------------------------------------------------------------------------------------------------------------------------------------------------------------------------------------------------------------------------------------------------------------------------------------------------------------------------------------------------------------------------------------------------------------------------------------------------------------------------------------------------------------------------------------------------------------------------------------------------------------------------------------------------------------------------------------------------------------------------------------------------------------------------------------------------------------------------------------------------------------------------------------------------------------------------------------------------------------------------------------------------------------------------------------------------------------------------------------------------------------------------------------------------------------------------------------------------------------------------------------------------------------------------------------------------------------------------------------------------------------------------------------------------------------------------------------------------------------------------------------------------------------------------------------------------------------------------------------------------------------------------------------------------------------------------------------------------------------------------------------------------------------------------------------------------------------------------------------------------------------------------------------------------------------|
| 109 | understanding of the mechanisms, regulating subtropical climate across the last interglacial, i.e., insolation,      |                                                                                                                                                                                                                                                                                                                                                                                                                                                                                                                                                                                                                                                                                                                                                                                                                                                                                                                                                                                                                                                                                                                                                                                                                                                                                                                                                                                                                                                                                                                                                                                                                                                                                                                                                                                                                                                                                                                                                                                                                                                                                                                                |
| 110 | oceanic and/or atmospheric forcing versus high-low-latitudes feedbacks,                                              |                                                                                                                                                                                                                                                                                                                                                                                                                                                                                                                                                                                                                                                                                                                                                                                                                                                                                                                                                                                                                                                                                                                                                                                                                                                                                                                                                                                                                                                                                                                                                                                                                                                                                                                                                                                                                                                                                                                                                                                                                                                                                                                                |
| 111 | Given its critical location near the origin of the Gulf Stream, sediments from downslope the shallow-water           |                                                                                                                                                                                                                                                                                                                                                                                                                                                                                                                                                                                                                                                                                                                                                                                                                                                                                                                                                                                                                                                                                                                                                                                                                                                                                                                                                                                                                                                                                                                                                                                                                                                                                                                                                                                                                                                                                                                                                                                                                                                                                                                                |
| 112 | carbonate platforms of the Bahamian archipelago (Fig. 1) have been previously investigated in terms of oceanic       | And and a state of the state of |
| 113 | and atmospheric variability (Slowey and Curry, 1995, Roth and Reijmer, 2004; 2005; Chabaud et al., 2016).            | and a second second                                                                                                                                                                                                                                                                                                                                                                                                                                                                                                                                                                                                                                                                                                                                                                                                                                                                                                                                                                                                                                                                                                                                                                                                                                                                                                                                                                                                                                                                                                                                                                                                                                                                                                                                                                                                                                                                                                                                                                                                                                                                                                            |
| 114 | However, a thorough study of the last interglacial climatic, evolution underpinned by a critical stratigraphical     | Althouse and a second                                                                                                                                                                                                                                                                                                                                                                                                                                                                                                                                                                                                                                                                                                                                                                                                                                                                                                                                                                                                                                                                                                                                                                                                                                                                                                                                                                                                                                                                                                                                                                                                                                                                                                                                                                                                                                                                                                                                                                                                                                                                                                          |
| 115 | insight is lacking so far. Here, a sediment record from the Little Bahama Bank (LBB) region is investigated for      |                                                                                                                                                                                                                                                                                                                                                                                                                                                                                                                                                                                                                                                                                                                                                                                                                                                                                                                                                                                                                                                                                                                                                                                                                                                                                                                                                                                                                                                                                                                                                                                                                                                                                                                                                                                                                                                                                                                                                                                                                                                                                                                                |
| 116 | possible links between the AMOC variability and the ITCZ during the last interglacial cycle. Today the LBB           |                                                                                                                                                                                                                                                                                                                                                                                                                                                                                                                                                                                                                                                                                                                                                                                                                                                                                                                                                                                                                                                                                                                                                                                                                                                                                                                                                                                                                                                                                                                                                                                                                                                                                                                                                                                                                                                                                                                                                                                                                                                                                                                                |
| 117 | region lies at the northern edge of the influence of the Atlantic Warm Pool, which expansion is strongly related     |                                                                                                                                                                                                                                                                                                                                                                                                                                                                                                                                                                                                                                                                                                                                                                                                                                                                                                                                                                                                                                                                                                                                                                                                                                                                                                                                                                                                                                                                                                                                                                                                                                                                                                                                                                                                                                                                                                                                                                                                                                                                                                                                |
| 118 | to the ITCZ movements (Wang and Lee, 2007; Levitus et al., 2013), making our site particularly sensitive to          |                                                                                                                                                                                                                                                                                                                                                                                                                                                                                                                                                                                                                                                                                                                                                                                                                                                                                                                                                                                                                                                                                                                                                                                                                                                                                                                                                                                                                                                                                                                                                                                                                                                                                                                                                                                                                                                                                                                                                                                                                                                                                                                                |
| 119 | monitor past shifts of the ITCZ. Given that geochemical properties of marine sediments around carbonate              |                                                                                                                                                                                                                                                                                                                                                                                                                                                                                                                                                                                                                                                                                                                                                                                                                                                                                                                                                                                                                                                                                                                                                                                                                                                                                                                                                                                                                                                                                                                                                                                                                                                                                                                                                                                                                                                                                                                                                                                                                                                                                                                                |
| 120 | platforms vary in response to sea level fluctuations (e.g., Lantzsch et al., 2007), X-ray fluorescense, (XRF) data   |                                                                                                                                                                                                                                                                                                                                                                                                                                                                                                                                                                                                                                                                                                                                                                                                                                                                                                                                                                                                                                                                                                                                                                                                                                                                                                                                                                                                                                                                                                                                                                                                                                                                                                                                                                                                                                                                                                                                                                                                                                                                                                                                |
| 121 | are being used together with stable isotope and faunal records to strengthen the temporal framework. Planktic        |                                                                                                                                                                                                                                                                                                                                                                                                                                                                                                                                                                                                                                                                                                                                                                                                                                                                                                                                                                                                                                                                                                                                                                                                                                                                                                                                                                                                                                                                                                                                                                                                                                                                                                                                                                                                                                                                                                                                                                                                                                                                                                                                |
| 122 | for a miniferal assemblage data complemented by $\delta^{18}O$ values, measured on surface- and thermocline-dwelling |                                                                                                                                                                                                                                                                                                                                                                                                                                                                                                                                                                                                                                                                                                                                                                                                                                                                                                                                                                                                                                                                                                                                                                                                                                                                                                                                                                                                                                                                                                                                                                                                                                                                                                                                                                                                                                                                                                                                                                                                                                                                                                                                |
| 123 | foraminifera, are employed to reconstruct the upper ocean properties (stratification, trends in temperature and      |                                                                                                                                                                                                                                                                                                                                                                                                                                                                                                                                                                                                                                                                                                                                                                                                                                                                                                                                                                                                                                                                                                                                                                                                                                                                                                                                                                                                                                                                                                                                                                                                                                                                                                                                                                                                                                                                                                                                                                                                                                                                                                                                |
| 124 | salinity), specifically looking at mechanisms controlling the foraminiferal assemblages. Assuming a coupling         | X                                                                                                                                                                                                                                                                                                                                                                                                                                                                                                                                                                                                                                                                                                                                                                                                                                                                                                                                                                                                                                                                                                                                                                                                                                                                                                                                                                                                                                                                                                                                                                                                                                                                                                                                                                                                                                                                                                                                                                                                                                                                                                                              |
| 125 | between foraminiferal assemblage data and past-mean annual positions of the ITCZ (Poore et al., 2003; Vautravers     | N                                                                                                                                                                                                                                                                                                                                                                                                                                                                                                                                                                                                                                                                                                                                                                                                                                                                                                                                                                                                                                                                                                                                                                                                                                                                                                                                                                                                                                                                                                                                                                                                                                                                                                                                                                                                                                                                                                                                                                                                                                                                                                                              |
| 126 | et al., 2007), our faunal records are then looked at in terms of potential geographical shifts of the ITCZ. Finally, |                                                                                                                                                                                                                                                                                                                                                                                                                                                                                                                                                                                                                                                                                                                                                                                                                                                                                                                                                                                                                                                                                                                                                                                                                                                                                                                                                                                                                                                                                                                                                                                                                                                                                                                                                                                                                                                                                                                                                                                                                                                                                                                                |
| 127 | we compare our new proxy records with published evidence from the regions of deep water formation, to draw           |                                                                                                                                                                                                                                                                                                                                                                                                                                                                                                                                                                                                                                                                                                                                                                                                                                                                                                                                                                                                                                                                                                                                                                                                                                                                                                                                                                                                                                                                                                                                                                                                                                                                                                                                                                                                                                                                                                                                                                                                                                                                                                                                |
| 128 | further conclusions on the subpolar forcing on the low-latitude climate during MIS 5e.                               | $\langle \rangle \rangle$                                                                                                                                                                                                                                                                                                                                                                                                                                                                                                                                                                                                                                                                                                                                                                                                                                                                                                                                                                                                                                                                                                                                                                                                                                                                                                                                                                                                                                                                                                                                                                                                                                                                                                                                                                                                                                                                                                                                                                                                                                                                                                      |
| 129 |                                                                                                                      |                                                                                                                                                                                                                                                                                                                                                                                                                                                                                                                                                                                                                                                                                                                                                                                                                                                                                                                                                                                                                                                                                                                                                                                                                                                                                                                                                                                                                                                                                                                                                                                                                                                                                                                                                                                                                                                                                                                                                                                                                                                                                                                                |
| 130 | 2 Regional Setting                                                                                                   |                                                                                                                                                                                                                                                                                                                                                                                                                                                                                                                                                                                                                                                                                                                                                                                                                                                                                                                                                                                                                                                                                                                                                                                                                                                                                                                                                                                                                                                                                                                                                                                                                                                                                                                                                                                                                                                                                                                                                                                                                                                                                                                                |
| 131 | 2.1 Hydrographic context                                                                                             |                                                                                                                                                                                                                                                                                                                                                                                                                                                                                                                                                                                                                                                                                                                                                                                                                                                                                                                                                                                                                                                                                                                                                                                                                                                                                                                                                                                                                                                                                                                                                                                                                                                                                                                                                                                                                                                                                                                                                                                                                                                                                                                                |
| 132 | Core MD99-2202 (27°34.5′ N, 78°57.9′ W, 460 m water depth) was taken from the upper northern slope of the            | $\in$                                                                                                                                                                                                                                                                                                                                                                                                                                                                                                                                                                                                                                                                                                                                                                                                                                                                                                                                                                                                                                                                                                                                                                                                                                                                                                                                                                                                                                                                                                                                                                                                                                                                                                                                                                                                                                                                                                                                                                                                                                                                                                                          |
| 133 | LBB, which is the northernmost shallow-water carbonate platform of the Bahamian archipelago. The study area          |                                                                                                                                                                                                                                                                                                                                                                                                                                                                                                                                                                                                                                                                                                                                                                                                                                                                                                                                                                                                                                                                                                                                                                                                                                                                                                                                                                                                                                                                                                                                                                                                                                                                                                                                                                                                                                                                                                                                                                                                                                                                                                                                |
| 134 | is at the western boundary of the wind-driven subtropical gyre (STG), in the vicinity to the Gulf Stream (Fig. 1a).  | $\leq$                                                                                                                                                                                                                                                                                                                                                                                                                                                                                                                                                                                                                                                                                                                                                                                                                                                                                                                                                                                                                                                                                                                                                                                                                                                                                                                                                                                                                                                                                                                                                                                                                                                                                                                                                                                                                                                                                                                                                                                                                                                                                                                         |
| 135 | The Gulf Stream supplies both heat and salt to the high northern latitudes thereby constituting the upper cell of    |                                                                                                                                                                                                                                                                                                                                                                                                                                                                                                                                                                                                                                                                                                                                                                                                                                                                                                                                                                                                                                                                                                                                                                                                                                                                                                                                                                                                                                                                                                                                                                                                                                                                                                                                                                                                                                                                                                                                                                                                                                                                                                                                |
| 136 | the AMOC,                                                                                                            | (                                                                                                                                                                                                                                                                                                                                                                                                                                                                                                                                                                                                                                                                                                                                                                                                                                                                                                                                                                                                                                                                                                                                                                                                                                                                                                                                                                                                                                                                                                                                                                                                                                                                                                                                                                                                                                                                                                                                                                                                                                                                                                                              |
| I   |                                                                                                                      | 1                                                                                                                                                                                                                                                                                                                                                                                                                                                                                                                                                                                                                                                                                                                                                                                                                                                                                                                                                                                                                                                                                                                                                                                                                                                                                                                                                                                                                                                                                                                                                                                                                                                                                                                                                                                                                                                                                                                                                                                                                                                                                                                              |

[revised manuscript text omitted]

eted: intertropical convergence zone (ITCZ

eted: ) (Fig. 1B-C)

**eted: a**

**leted:** pool of waters ( $T > 28^{\circ}$ C) which expands into the nama region from the Caribbean Sea and the equatorial antic (Stramma and Schott, 1999; Wang and Lee, 2007; ritus et al., 2013). Today, the LBB region lies at the thern edge of the influence of tropical pool waters, making site particularly sensitive to monitor past shifts of the CZ.

| Deleted: a      |  |
|-----------------|--|
| Deleted: layer  |  |
| Deleted: 12<    |  |
| Deleted:        |  |
| Deleted: driven |  |
| Deleted: from   |  |
| Deleted: s      |  |

| -( | Deleted: | Typical                |
|----|----------|------------------------|
| -( | Deleted: | As in a                |
| (  | Deleted: | for                    |
| -( | Deleted: | as well as by          |
| -( | Deleted: | ; Chabaud et al., 2016 |
| (  | Deleted: | whereas                |
| (  | Deleted: | are                    |
| X  | Deleted: | d                      |
| C  | Deleted: | producing              |
| X  | Deleted: | high resolution        |
| Y  | Deleted: | interglacial climate   |

| 283        | consolidated sediments are formed from the pelagic organisms (Droxler and Schlager, 1985; Slowey et al., 2002;                               |              |                                                                                                                                        |
|------------|----------------------------------------------------------------------------------------------------------------------------------------------|--------------|----------------------------------------------------------------------------------------------------------------------------------------|
| 284
bes | Lantzsch et al., 2007).                                                                                                                      |              |                                                                                                                                        |
| 285        |                                                                                                                                              |              |                                                                                                                                        |
| 286        | 3 Methods                                                                                                                                    |              | Deleted: 1                                                                                                                             |
| 287        | 3.1 Foraminiferal counts and stable isotopes analyses                                                                                        |              |                                                                                                                                        |
| 288        | Planktic foraminiferal assemblages were counted on representative splits of the 150-250 µm fraction containing                               |              | Deleted: i                                                                                                                             |
| 289        | at least 300 individual specimens. Counts were also performed in the ${>}250\mu\text{m}$ fraction. The census data from the                  |              |                                                                                                                                        |
| 290        | two size fractions were added up and recalculated into relative abundance of planktic foraminifera in the fraction                           |              |                                                                                                                                        |
| 291        | ${>}150~\mu\text{m}.$ Faunal data were obtained at each 2 cm for the core section between 508.5 and 244.5 cm and at each                     |              |                                                                                                                                        |
| 292        | 10 cm between 240.5 and 150.5 cm. According to a standard practice, Globorotalia menardii and Globorotalia                                   |              |                                                                                                                                        |
| 293        | tumida as well as Globigerinoides sacculifer and Globigerinoides trilobus were grouped together, and referred to                             |              | Deleted: (Poore et al., 2003; Kandiano et al., 2012;                                                                            |
| 294        | as G. menardii and G. sacculifer, respectively (Poore et al., 2003; Kandiano et al., 2012; Jentzen et al., 2018).                            |              | Chabaud, 2016)                                                                                                                         |
| 295        | New oxygen isotope data were produced at 2 cm steps using ~10-30 tests of Globorotalia truncatulinoides (dex)                                |              |                                                                                                                                        |
| 296        | and ~5-20 tests of Globorotalia inflata for depths 508.5-244.5 cm and 508.5-420.5 cm, respectively. Analyses                                 |              |                                                                                                                                        |
| 297        | were performed using a Finnigan MAT 253 mass spectrometer at the GEOMAR Stable Isotope Laboratory.                                           |              |                                                                                                                                        |
| 298        | Calibration to the Vienna Pee Dee Belemnite (VPDB) isotope scale was made via the NBS-19 and an internal                                     |              |                                                                                                                                        |
| 299        | laboratory standard. The analytical precision of in-house standards was better than 0.07‰ (1 $\sigma$ ) for $\delta^{18}$ O. Isotopic |              |                                                                                                                                        |
| 300        | data derived from the deep-dwelling foraminifera G. truncatulinoides (dex) and G. inflata could be largely                                   |              |                                                                                                                                        |
| 301        | associated with the permanent thermocline and linked to winter conditions (Groeneveld and Chiessi, 2011;                                     |              |                                                                                                                                        |
| 302        | Jonkers and Kučera, 2017; Jentzen et al., 2018). However, as calcification of their tests starts already in the mixed                        |              | Deleted: Deep-dwelling foraminifera G. truncatulinoides                                                                                |
| 303        | layer and continues in the main thermocline (Fig. 1c), the abovementioned species are thought to accumulate in                               |              | and G. inflata are found in greatest abundances at the base of
the seasonal thermocline (100-200 m), under environmental     |
| 304        | their tests hydrographic signals from different water depths (Groeneveld and Chiessi, 2011; Mulitza et al., 1997).                           |              | species can migrate to greater depths (Cléroux et al., 2007). A                                                                        |
| 305        |                                                                                                                                              |              | Deleted: ing                                                                                                                           |
| 306        | 3.2 XRF scanning                                                                                                                             |              | Deleted: Also, isotopic data derived from G .                                                                            |
| 307        | XRF analysis was performed in two different runs using the Aavatech XRF Core Scanner at Christian-Albrecht                                   |              | truncatulinoides and G. inflata bear a cold-season weighted signal, as these species are abundant in the N. Atlantic STG |
| 308        | University of Kiel (for technical details see Richter et al., 2006). To obtain intensities of elements with lower                            | $\mathbb{N}$ | during winter-spring time (Jonkers and Kucera, 2015).                                                                                  |
| 309        | atomic weight (e.g., calcium (Ca), chlorine (Cl)), XRF scanning measurements were carried out with the X-ray                                 |              | Deleted: X-ray fluorescence (                                                                                                          |
| 310        | tube voltage of 10 kv, the tube current of 750 $\mu$ A and the counting time of 10 seconds. To analyze heavy elements                        | /            | Deleted:                                                                                                                               |
| 311        | (e.g., iron (Fe), Sr), the X-ray generator setting of 30 kv and 2000 $\mu$ A and the counting time of 20 seconds were                        |              |                                                                                                                                        |
| 312        | used; a palladium thick filter was placed in the X-ray tube to reduce the high background radiation generated by                             |              |                                                                                                                                        |

[revised manuscript text omitted]

| 376 | for
aminiferal $\delta^{18}O$ record at 456 cm, with the onset of MIS 5e "plateau" at ~129 ka (Masson-Delmotte et al.,                                                                                                                                                                                                                                                                                                                                                                                                                                                                                                                                                                                                                                                                                                                                                                                                                                                                                                                                                                                                                                                                                                                                                                                                                                                                                                                                                                                                                                                                                                                                                                                                                                                                                                                                                                                                                                                                                                                                                                                                      |                       |                                                                                                                              |
|-----|--------------------------------------------------------------------------------------------------------------------------------------------------------------------------------------------------------------------------------------------------------------------------------------------------------------------------------------------------------------------------------------------------------------------------------------------------------------------------------------------------------------------------------------------------------------------------------------------------------------------------------------------------------------------------------------------------------------------------------------------------------------------------------------------------------------------------------------------------------------------------------------------------------------------------------------------------------------------------------------------------------------------------------------------------------------------------------------------------------------------------------------------------------------------------------------------------------------------------------------------------------------------------------------------------------------------------------------------------------------------------------------------------------------------------------------------------------------------------------------------------------------------------------------------------------------------------------------------------------------------------------------------------------------------------------------------------------------------------------------------------------------------------------------------------------------------------------------------------------------------------------------------------------------------------------------------------------------------------------------------------------------------------------------------------------------------------------------------------------------------------------|-----------------------|------------------------------------------------------------------------------------------------------------------------------|
| 377 | 2013). This age is in good agreement with many marine and speleothem records, dating a rapid post-stadial                                                                                                                                                                                                                                                                                                                                                                                                                                                                                                                                                                                                                                                                                                                                                                                                                                                                                                                                                                                                                                                                                                                                                                                                                                                                                                                                                                                                                                                                                                                                                                                                                                                                                                                                                                                                                                                                                                                                                                                                                      |                       |                                                                                                                              |
| 378 | warming and monsoon intensification to 129-128.7 ka (Govin et al., 2015; Jiménez-Amat and Zahn, 2015; Deaney                                                                                                                                                                                                                                                                                                                                                                                                                                                                                                                                                                                                                                                                                                                                                                                                                                                                                                                                                                                                                                                                                                                                                                                                                                                                                                                                                                                                                                                                                                                                                                                                                                                                                                                                                                                                                                                                                                                                                                                                                   |                       | (Deleted: Galaasen et al., 2014;                                                                                             |
| 379 | et al., 2017), coincident with the sharp methane increase in the EPICA Dome C ice core (Loulergue et al., 2008;                                                                                                                                                                                                                                                                                                                                                                                                                                                                                                                                                                                                                                                                                                                                                                                                                                                                                                                                                                                                                                                                                                                                                                                                                                                                                                                                                                                                                                                                                                                                                                                                                                                                                                                                                                                                                                                                                                                                                                                                                |                       |                                                                                                                              |
| 380 | Govin et al., 2012). Although we do not apply a specific age marker to frame the decline of the MIS 5e "plateau",                                                                                                                                                                                                                                                                                                                                                                                                                                                                                                                                                                                                                                                                                                                                                                                                                                                                                                                                                                                                                                                                                                                                                                                                                                                                                                                                                                                                                                                                                                                                                                                                                                                                                                                                                                                                                                                                                                                                                                                                              |                       |                                                                                                                              |
| 381 | the resulting decrease in the percentage of warm surface-dwelling foraminifera of Globigerinoides genus as well                                                                                                                                                                                                                                                                                                                                                                                                                                                                                                                                                                                                                                                                                                                                                                                                                                                                                                                                                                                                                                                                                                                                                                                                                                                                                                                                                                                                                                                                                                                                                                                                                                                                                                                                                                                                                                                                                                                                                                                                                |                       |                                                                                                                              |
| 382 | as the initial increase in the planktic $\delta^{18}$ O values dates back to ~117 ka (Figs. 3-5), which broadly coincides with                                                                                                                                                                                                                                                                                                                                                                                                                                                                                                                                                                                                                                                                                                                                                                                                                                                                                                                                                                                                                                                                                                                                                                                                                                                                                                                                                                                                                                                                                                                                                                                                                                                                                                                                                                                                                                                                                                                                                                                                 |                       | (Deleted: 4                                                                                                                  |
| 383 | the cooling over Greenland (NGRIP community members, 2004). A similar subtropical-polar climatic coupling                                                                                                                                                                                                                                                                                                                                                                                                                                                                                                                                                                                                                                                                                                                                                                                                                                                                                                                                                                                                                                                                                                                                                                                                                                                                                                                                                                                                                                                                                                                                                                                                                                                                                                                                                                                                                                                                                                                                                                                                                      |                       |                                                                                                                              |
| 384 | was proposed in earlier studies from the western North, Atlantic STG (e.g., Vautravers et al., 2004; Schmidt et al.,                                                                                                                                                                                                                                                                                                                                                                                                                                                                                                                                                                                                                                                                                                                                                                                                                                                                                                                                                                                                                                                                                                                                                                                                                                                                                                                                                                                                                                                                                                                                                                                                                                                                                                                                                                                                                                                                                                                                                                                                           |                       | Deleted:                                                                                                                     |
| 385 | 2006a; Bahr et al., 2013; Deaney et al., 2017).                                                                                                                                                                                                                                                                                                                                                                                                                                                                                                                                                                                                                                                                                                                                                                                                                                                                                                                                                                                                                                                                                                                                                                                                                                                                                                                                                                                                                                                                                                                                                                                                                                                                                                                                                                                                                                                                                                                                                                                                                                                                                |                       |                                                                                                                              |
| 386 |                                                                                                                                                                                                                                                                                                                                                                                                                                                                                                                                                                                                                                                                                                                                                                                                                                                                                                                                                                                                                                                                                                                                                                                                                                                                                                                                                                                                                                                                                                                                                                                                                                                                                                                                                                                                                                                                                                                                                                                                                                                                                                                                |                       |                                                                                                                              |
| 387 | 5 Results                                                                                                                                                                                                                                                                                                                                                                                                                                                                                                                                                                                                                                                                                                                                                                                                                                                                                                                                                                                                                                                                                                                                                                                                                                                                                                                                                                                                                                                                                                                                                                                                                                                                                                                                                                                                                                                                                                                                                                                                                                                                                                                      |                       |                                                                                                                              |
| 388 | 5.1 XRF data in the lithological context                                                                                                                                                                                                                                                                                                                                                                                                                                                                                                                                                                                                                                                                                                                                                                                                                                                                                                                                                                                                                                                                                                                                                                                                                                                                                                                                                                                                                                                                                                                                                                                                                                                                                                                                                                                                                                                                                                                                                                                                                                                                                       |                       |                                                                                                                              |
| 389 | In Fig. 3. XRF-derived elemental data are plotted against lithological and sedimentological records. Beyond the                                                                                                                                                                                                                                                                                                                                                                                                                                                                                                                                                                                                                                                                                                                                                                                                                                                                                                                                                                                                                                                                                                                                                                                                                                                                                                                                                                                                                                                                                                                                                                                                                                                                                                                                                                                                                                                                                                                                                                                                                |                       | (Deleted: 2                                                                                                                  |
| 390 | intervals with low Ca counts and correspondingly high Cl intensities (at 300-325 cm and 395-440 cm). Ca                                                                                                                                                                                                                                                                                                                                                                                                                                                                                                                                                                                                                                                                                                                                                                                                                                                                                                                                                                                                                                                                                                                                                                                                                                                                                                                                                                                                                                                                                                                                                                                                                                                                                                                                                                                                                                                                                                                                                                                                                        |                       | Deleted: physical                                                                                                            |
| 201 |                                                                                                                                                                                                                                                                                                                                                                                                                                                                                                                                                                                                                                                                                                                                                                                                                                                                                                                                                                                                                                                                                                                                                                                                                                                                                                                                                                                                                                                                                                                                                                                                                                                                                                                                                                                                                                                                                                                                                                                                                                                                                                                                |                       | Deleted: properties                                                                                                          |
| 391 | intensities do not vary significantly, which is in line with a stable carbonate content of about 94.% Wt (Lantzsch                                                                                                                                                                                                                                                                                                                                                                                                                                                                                                                                                                                                                                                                                                                                                                                                                                                                                                                                                                                                                                                                                                                                                                                                                                                                                                                                                                                                                                                                                                                                                                                                                                                                                                                                                                                                                                                                                                                                                                                                             |                       | Deleted: to                                                                                                                  |
| 392 | et al., 2007). Our Sr record closely follows the aragonite curve, demonstrating that the interglacial minerology is                                                                                                                                                                                                                                                                                                                                                                                                                                                                                                                                                                                                                                                                                                                                                                                                                                                                                                                                                                                                                                                                                                                                                                                                                                                                                                                                                                                                                                                                                                                                                                                                                                                                                                                                                                                                                                                                                                                                                                                                            |                       | (Deleted: and the grain size data                                                                                            |
| 393 | dominated by aragonite. Beyond the intervals containing reduced Ca intensities, a good coherence between Sr/Ca                                                                                                                                                                                                                                                                                                                                                                                                                                                                                                                                                                                                                                                                                                                                                                                                                                                                                                                                                                                                                                                                                                                                                                                                                                                                                                                                                                                                                                                                                                                                                                                                                                                                                                                                                                                                                                                                                                                                                                                                                 |                       |                                                                                                                              |
| 394 | and aragonite content is observed. The rapid increase in Sr/Ca and aragonite is found at the end of the penultimate                                                                                                                                                                                                                                                                                                                                                                                                                                                                                                                                                                                                                                                                                                                                                                                                                                                                                                                                                                                                                                                                                                                                                                                                                                                                                                                                                                                                                                                                                                                                                                                                                                                                                                                                                                                                                                                                                                                                                                                                            |                       |                                                                                                                              |
| 395 | deglaciation (T2), coeval with the elevated absolute abundances of G. menardii per sample (Fig. 3). The gradual                                                                                                                                                                                                                                                                                                                                                                                                                                                                                                                                                                                                                                                                                                                                                                                                                                                                                                                                                                                                                                                                                                                                                                                                                                                                                                                                                                                                                                                                                                                                                                                                                                                                                                                                                                                                                                                                                                                                                                                                                | ~                     | Deleted: Termination 2,                                                                                                      |
| 396 | step-like Sr/Ca and aragonite decrease characterizes both the glacial inception and the later MIS 5 phase.                                                                                                                                                                                                                                                                                                                                                                                                                                                                                                                                                                                                                                                                                                                                                                                                                                                                                                                                                                                                                                                                                                                                                                                                                                                                                                                                                                                                                                                                                                                                                                                                                                                                                                                                                                                                                                                                                                                                                                                                                     |                       | (Deleted: 4                                                                                                                  |
| 397 | Intensities of Fe abruptly decrease at the beginning of the last interglacial, but gradually increase during the glacial                                                                                                                                                                                                                                                                                                                                                                                                                                                                                                                                                                                                                                                                                                                                                                                                                                                                                                                                                                                                                                                                                                                                                                                                                                                                                                                                                                                                                                                                                                                                                                                                                                                                                                                                                                                                                                                                                                                                                                                                       |                       |                                                                                                                              |
| 398 | inception (Fig. 4). Between ~112 and 114.5 ka, the actual XRF measurements were affected by a low sediment                                                                                                                                                                                                                                                                                                                                                                                                                                                                                                                                                                                                                                                                                                                                                                                                                                                                                                                                                                                                                                                                                                                                                                                                                                                                                                                                                                                                                                                                                                                                                                                                                                                                                                                                                                                                                                                                                                                                                                                                                     |                       | Deleted: 5D                                                                                                                  |
| 399 | level in the core tube.

---

## Author Response (AR2)

**Author´s response to the comments of the Anonymous Referee #2 (minor revision from 20 August 2018)**

**Please note than the line numbers are referred to the mark up version below.**

Zhuravleva and Bauch "Last Interglacial ocean changes in the Bahamas: climate and teleconnections between the low and high latitudes".

The revised manuscript is much improved, with a much sharper focus, discussion (including mechanisms, as requested by both reviewers) and conclusions. The paper provides an excellent record of the evolution of hydrographic conditions at the Little Bahama Bank (LBB) for Termination II (TII) and the Last Interglacial (LIg), reflecting both the insolation driven and AMOC modulated migration of the ITCZ.

The paper will make a good contribution to the journal, providing high-resolution evidence of teleconnections between the low and high latitudes for TII and the LIg. Based on the revised manuscript, I would recommend publication subject to some minor revisions/technical corrections.

Suggestions for revision or reasons for rejection (will be published if the paper is accepted for final publication)

Comments:

Platform sedimentology and sea level: This section is much improved. However, I would consider removing lines 251 to 256, this really doesn't add anything. You wouldn't expect to be able to resolve intra-LIg sea level variations from your data.

**Lines 251-256 removed with the exception of one sentence (lines 267-274)**

line 259: consider using "elevated" rather than "high proportions" – the percentages are still very low (<10% and < 2% for G. inflata and G. truncatulinoides (dex.), respectively).

**Done (line 291)**

line 280: change "reversion" (in what???) to "oscillation" and clarify what you are referring to (surface hydrographic conditions?)

**Done. "Reversion" changed to "oscillation" (line 333). "Past hydrographic conditions" changed to "past fluctuations in seawater temperature and salinity" (lines 332-333).**

Mechanisms influencing changing faunal abundances: Reviewer 1 requested clarification that water column stratification is not the only influence upon the abundance of G. truncatulinoides (dex.). This has not been addressed by the authors (lines 293 to 301). I agree with the authors interpretation, however, there should be an acknowledgment of alternative explanations. This could be easily fixed in line 293/294 with brackets.

**We agree with the Reviewer´s comment on the variety of mechanisms that can influence the thermocline-associated assemblage and point it out in lines 346-347. Furthermore, we do highlight some of the mechanisms (i.e., seasonal variation in salinity, temperature) in the following lines 347-351.**

Use of "Younger Dryas-type event": I am averse to this terminology, given the very different 'background states' for the two events. This is a personal view and the authors do highlight this (line 379 to 381).

**Although we agree with the Reviewer´s comment on different background conditions underlying the Younger Dryas and the climatic event at 127 ka, we refer to the pronounced millennial-scale cooling/salinification event as to a Younger Dryas – like event, also because it was used in earlier studies (as stated in line 437), e.g., Sarnthein and Tiedemann (1990), Bauch et al. (2012) or Jiménez-Amat and Zahn (2015). By doing this, we acknowledge its comparable stratigraphical positioning and climatic significance within a deglacial termination in general.**

Technical corrections:

line 217: change "unstable" to "variable".

**Done (line 248).**

Figures

1. Figure captions for figures 3 to 6 – please state what the dashed vertical lines are.

**Done.**

2. Figure 4: I found this rather hard to read. Could you use other colours other than black and blue?

**Done. Blue color was changed to magenta.**

3. Figure 4: either the vertical dashed line or the shaded blue bar at 131 ka is not vertical – please fix.

**Fixed.**

4. Figure 4: the max. tick marks for the vertical axis for % G. inflata is missing – please add.

**Done.**

5. Figure 4: Please add what the vertical blue bars indicate to the caption.

**Done.**

6. Figure 5: consider adding the blue bars of the stratification/cooling events at 131 ka and 127 ka

**Figure 5 deals with the mixed layer properties and is referred to only in the chapter 6.3, focusing on climatic mechanisms influencing the subtropical climate, while the stratification events are discussed in the chapter 6.2 and demonstrated in Figure 4. Thus, we refrain from highlighting the stratification events in Figure 5, as this would overburden the figure by including unnecessary information.**

[revised manuscript text omitted]